# Genome-wide identification of DNA methylation QTLs in whole blood highlights pathways for cardiovascular disease

Tianxiao Huan[1,2,15], Roby Joehanes[1,2,15], Ci Song[1,2,3,4], Fen Peng[5], Yichen Guo [6,7], Michael Mendelson[1,2,8], Chen Yao [1,2], Chunyu Liu[9], Jiantao Ma[1,2], Melissa Richard [5], Golareh Agha[10], Weihua Guan[11], Lynn M. Almli [12], Karen N. Conneely[13], Joshua Keefe[1,2], Shih-Jen Hwang[1,2], Andrew D. Johnson[1,2], Myriam Fornage[5], Liming Liang [7,14] & Daniel Levy[1,2]

Identifying methylation quantitative trait loci (meQTLs) and integrating them with disease-associated variants from genome-wide association studies (GWAS) may illuminate functional mechanisms underlying genetic variant-disease associations. Here, we perform GWAS of >415 thousand CpG methylation sites in whole blood from 4170 individuals and map 4.7 million *cis*- and 630 thousand *trans*-meQTL variants targeting >120 thousand CpGs. Independent replication is performed in 1347 participants from two studies. By linking *cis*-meQTL variants with GWAS results for cardiovascular disease (CVD) traits, we identify 92 putatively causal CpGs for CVD traits by Mendelian randomization analysis. Further integrating gene expression data reveals evidence of *cis* CpG-transcript pairs causally linked to CVD. In addition, we identify 22 *trans*-meQTL hotspots each targeting more than 30 CpGs and find that *trans*-meQTL hotspots appear to act in *cis* on expression of nearby transcriptional regulatory genes. Our findings provide a powerful meQTL resource and shed light on DNA methylation involvement in human diseases.

[1] The Framingham Heart Study, Framingham, MA, USA. [2] The Population Sciences Branch, Division of Intramural Research, National Heart, Lung, and Blood Institute, National Institutes of Health, Bethesda, MD, USA. [3] Department of Medical Sciences, Uppsala University, Uppsala, Sweden. [4] Department of Immunology, Genetics and Pathology, Uppsala University, Uppsala, Sweden. [5] Brown Foundation Institute of Molecular Medicine, McGovern Medical School, University of Texas Health Science Center at Houston, Houston, TX, USA. [6] Department of Environmental Health, Harvard T.H. Chan School of Public Health, Harvard University, Boston, MA, USA. [7] Department of Biostatistics, Harvard T.H. Chan School of Public Health, Harvard University, Boston, MA, USA. [8] Department of Cardiology, Boston Children's Hospital, Harvard University, Boston, MA, USA. [9] Department of Biostatistics, Boston University School of Public Health, Boston, MA, USA. [10] Mailman School of Public Health, Columbia University, New York City, NY, USA. [11] Division of Biostatistics, School of Public Health, University of Minnesota, Minneapolis, MN, USA. [12] Department of Psychiatry and Behavioral Sciences, Emory University School of Medicine, Atlanta, GA, USA. [13] Department of Human Genetics, Emory University School of Medicine, Atlanta, GA, USA. [14] Department of Epidemiology, Harvard T.H. Chan School of Public Health, Boston, MA, USA. [15]These authors contributed equally:Tianxiao Huan, Roby Joehanes. Correspondence and requests for materials should be addressed to T.H. (email: tianxiao.huan@nih.gov) or to L.L. (email: lliang@hsph.harvard.edu) or to D.L. (email: Levyd@nih.gov)

DNA methylation (DNAm), the covalent binding of a methyl group to the 5' carbon of cytosine occurring mainly at CpG dinucleotide sequences in the genome, is an important epigenetic regulatory mechanism, and plays a critical role in the regulation of gene expression[1]. Site-specific DNAm is associated with many complex human diseases and traits[2]. DNAm variability is genetically influenced[3,4], accrues with human aging[5], and can be altered by environmental exposures such as smoking[6] and alcohol consumption[7]. Epigenome-wide association studies (EWAS) have identified differentially methylated CpGs associated with numerous clinical traits. The strongest CpGs identified by EWAS, however, seldom reflect a causal role in disease (i.e., CpG → disease), but rather reflect downstream effects of disease processes on the methylome (i.e., disease → CpG)[8,9]. Nevertheless, CpGs can serve as useful biomarkers of disease, and the identification of a subset of CpGs that have causal roles in disease can provide insights into disease etiology and potential therapeutic targets.

Genome-wide association studies (GWAS) have recently identified genetic loci associated with site-specific DNAm of CpGs, known as DNA methylation quantitative trait loci (meQTLs)[10–14]. We hypothesized that identifying meQTL variants and linking them to disease-associated genetic variants from GWAS would pinpoint molecular mechanisms underlying genetic susceptibility to human diseases that are due, at least in part, to altered epigenetic regulation. Additionally, it could help explain the molecular consequences of non-protein-coding, disease-associated genetic variants from GWAS.

To this end, we perform genome-wide association testing of genetic variants with whole blood DNAm from 4170 European ancestry (EA) participants in the Framingham Heart Study (FHS) and comprehensively map *cis*- and *trans*-meQTLs. External replication is performed in 963 EA participants in the Atherosclerosis Risk in Communities (ARIC) study and 384 African-American ancestry (AA) participants in the Grady Trauma Project (GTP). We link *cis*-meQTL single nucleotide polymorphisms (SNPs) with GWAS results for cardiovascular disease (CVD) and its metabolic risk factors and employ Mendelian randomization (MR) to identify putatively causal CpGs for CVD and its risk factors. We further integrate gene expression, gene expression-associated CpGs, and gene expression-associated QTLs (eQTLs), to reveal causal genomic regulatory pathways for CVD traits. We also report *trans*-meQTL hotspots, each targeting 30 or more CpGs and demonstrate their influence on *cis* transcriptional regulatory genes.

## Results

### Heritability of global DNAm in peripheral blood.
The clinical characteristics of the 4170 FHS participants in this study are summarized in Supplementary Table 1, including 2648 participants from the FHS offspring cohort (mean age 66 years, 54% women), and 1522 participants from the FHS third generation cohort (mean age 45 years, 52% women). Among the study participants, 456 are unrelated individuals, and 3,714 are from 511 families of varying sizes (Supplementary Table 2). Figure 1a displays the distribution of heritability estimates ($h^2_{CpG}$) for 415,318 CpGs. The average heritability of DNAm across all CpGs was estimated to be 0.09 ± 0.02 (mean ± SD) and 105,622 CpGs (25.4%) were found to have $h^2_{CpG}$ > 0.1 (Supplementary Data 1), 39,090 (9.4%) have $h^2_{CpG}$ > 0.3, and 5416 (1.3%) have $h^2_{CpG}$ > 0.6. Genomic features enrichment analysis revealed that CpGs with $h^2_{CpG}$ > 0.1 were highly enriched for location in enhancer regions (fold enrichment = 1.24, $P < 1E-16$, hypergeometric test), and depleted in promoter, 0–200 bases upstream of transcription start

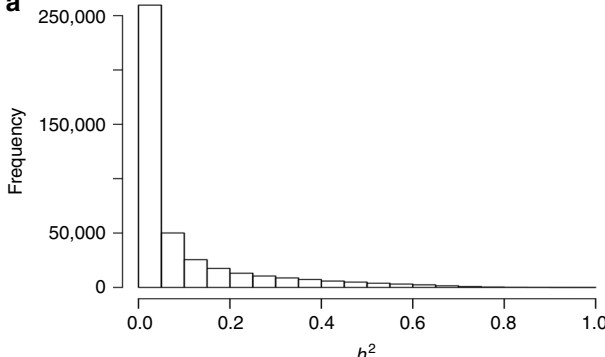

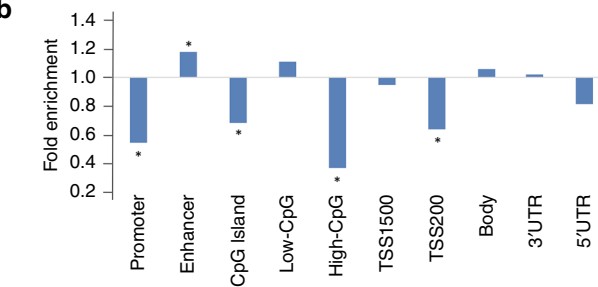

**Fig. 1** Heritability analysis of CpGs genome-wide. **a** Heritability ($h^2_{CpG}$) distribution of 415,318 CpGs; **b** enrichment of CpGs with $h^2_{CpG} > 0.1$ in different genomic regions

sites (TSS200), CpG island, and high-CpG dense regions (fold enrichment < 0.8, $P < 1E-16$, hypergeometric test, Fig. 1b). Household effects were positively associated with $h^2_{CpG}$, with Pearson correlation $r = 0.44$. There were 6,212 CpGs with household effects >0.1 (Supplementary Data 2) that showed enrichment for location in 3'UTR regions (fold enrichment = 1.26, $P < 1E-16$, hypergeometric test) and depletion in promoter, TSS200, CpG Islands, and high-CpG dense regions (fold enrichment <0.8, $P < 1E-16$, hypergeometric test, Supplementary Fig 1).

### Identification and replication of *cis*- and *trans*-meQTLs.
Pairwise association analyses were performed for 8.5 million SNPs and 415 thousand CpGs measured in whole blood samples from 4170 FHS participants. *cis*-meQTLs were defined as SNPs residing within 1 Mb upstream or downstream of a CpG site. The distribution of *cis*-meQTLs in relation to distance from the corresponding CpGs suggested that a 2 Mb window is a reasonable window for mapping *cis*-meQTLs (Supplementary Fig 2). We identified 4.7 million *cis*- (for 121.6 thousand CpGs, 26.8 million pairs) and 706 thousand *trans*-meQTL SNPs (for 13.5 thousand CpGs, 2 million pairs) at Bonferroni-corrected $P < 0.05$ ($P < 2E-11$ for *cis* associations for $2.5 \times 10^9$ pairs, and $P < 1.5E-14$ for *trans* associations for $3.5 \times 10^{12}$ pairs). Among *trans*-meQTL SNPs, 33% reside on the same chromosome as the corresponding CpGs, i.e., intrachromosomal *trans*-meQTLs, and 67% were interchromosomal *trans*-meQTLs. Among intrachromosomal *trans*-meQTLs SNPs, 70% reside within 5 Mb of the corresponding CpGs. This indicates that many presumed intrachromosomal *trans*-meQTLs are not true *trans*-meQTLs but instead long-range *cis*-meQTLs. We, therefore, excluded long-range *cis*-meQTLs (i.e., SNPs > 1 Mb but within 5 Mb of CpGs), leaving 630 thousand *trans*-meQTL SNPs (for 10.6 thousand CpGs, 1.6 million pairs). After pruning redundant SNPs at a given locus by limiting linkage disequilibrium [LD] to $r^2 < 0.2$, there remained

394 thousand independent *cis*- and 21 thousand *trans*-meQTL SNPs.

Our study, with the largest sample size in a single-site meQTL investigation to date, provides obvious benefits in terms of greater statistical power for discovery (Supplementary Fig 3 displays a flowchart for the identification and replication of meQTLs). We did not observe inflation of the genomic control factor ($\lambda = 0.93$, Supplementary Fig 4). By overlapping our meQTL SNP-CpG pairs with previously published results identified in whole blood[10,11,13] (Supplementary Fig 5), we estimated that our meQTL SNP-CpG pairs cover 80–90% of significant meQTL SNP-CpG pairs reported in two prior studies[10,13]. Because of the much larger sample size in our study, we detected 3.5 times more *cis*- and 10 times more *trans*-meQTL SNPs. A total of 6.9 million *cis*-meQTL SNP-CpG pairs (27%) and 206 thousand *trans*- pairs (13%) identified in our study were reported in at least one of the previous meQTL studies.

We further validated our meQTLs by performing independent external replication analysis based on 963 EA samples from the ARIC study and 384 AA samples from GTP. We found that more than 99% of *cis*- and *trans*-meQTL SNP-CpG pairs showed consistent allelic direction of effect in relation to methylation in ARIC vs. FHS (Fig. 2a, b), and 81% of *cis*- and *trans*- pairs showed consistent directions of effect in GTP vs. FHS (Fig. 2c, d). Thirty-six percent of *cis*- and 39% of *trans*-meQTL SNP-CpG pairs replicated at Bonferroni-corrected $P < 0.05$ (corrected for the 26.8 million *cis*- pairs and 2 million *trans*- pairs), and 91% of *cis* and 94% of *trans* pairs replicated at nominal $P < 0.05$ in ARIC. 10% of *cis*- and 6% of *trans*- meQTL SNP-CpG pairs from FHS replicated at Bonferroni-corrected

$P < 0.05$, and 51% of *cis*- and 41% of *trans*- pairs replicated at nominal $P < 0.05$ in GTP.

**Characteristics of *cis*- and *trans*-meQTLs.** Figure 3a shows that CpGs with higher heritability ($h^2_{CpG}$) are more likely to be associated with *cis*- and *trans*-meQTL SNPs. Among the CpGs with $h^2_{CpG} > 0.1$, 73% have at least one *cis*-meQTL SNP, and 7% have at least one *trans*-meQTL SNP. Among the CpGs with $h^2_{CpG} > 0.6$, 76% have at least one *cis*-meQTL SNP, and 8% have at least one *trans*-meQTL SNP. The mean (±SD) proportions of inter-individual variation in CpGs explained by the most significant single *cis*-meQTL SNP ($h^2_{cis-meQTL}$) is $0.08 \pm 0.10$ and for the most significant single *trans*-meQTL SNP ($h^2_{trans-meQTL}$) it is $0.05 \pm 0.06$. The proportions of inter-individual variation in CpGs explained by all of its corresponding *cis*-meQTL SNPs in aggregate ($h^2_{cis-meQTLs}$) is $0.22 \pm 0.25$ and by all of its corresponding *trans*-meQTL SNPs ($h^2_{trans-meQTLs}$) is $0.18 \pm 0.21$ (Fig. 3b). $h^2_{cis-meQTLs}$ and $h^2_{trans-meQTLs}$ are positively correlated with $h^2_{CpG}$, with Pearson correlations $r = 0.57$ and $r = 0.25$, respectively (Fig. 3c). For some meQTLs, $h^2_{meQTL}$ estimates were large. For example, $h^2_{meQTL}$ for the *cis*-meQTL-CpG pair rs62396312–cg03644281 is 0.6. rs62396312 with minor allele frequency (MAF) = 0.11 is 23 Kb from cg03644281, which is in the 3'UTR of *NFYA* (Fig. 3d). $h^2_{meQTL}$ for *trans*-meQTL-CpG pair rs2296406–cg04657470 is 0.57. rs2296406 (MAF = 0.38) is on Chromosome 16, and cg04657470 is in the first exon of *HSPE1* and the 5'UTR of *HSPD1* on Chromosome 2 (Fig. 3e).

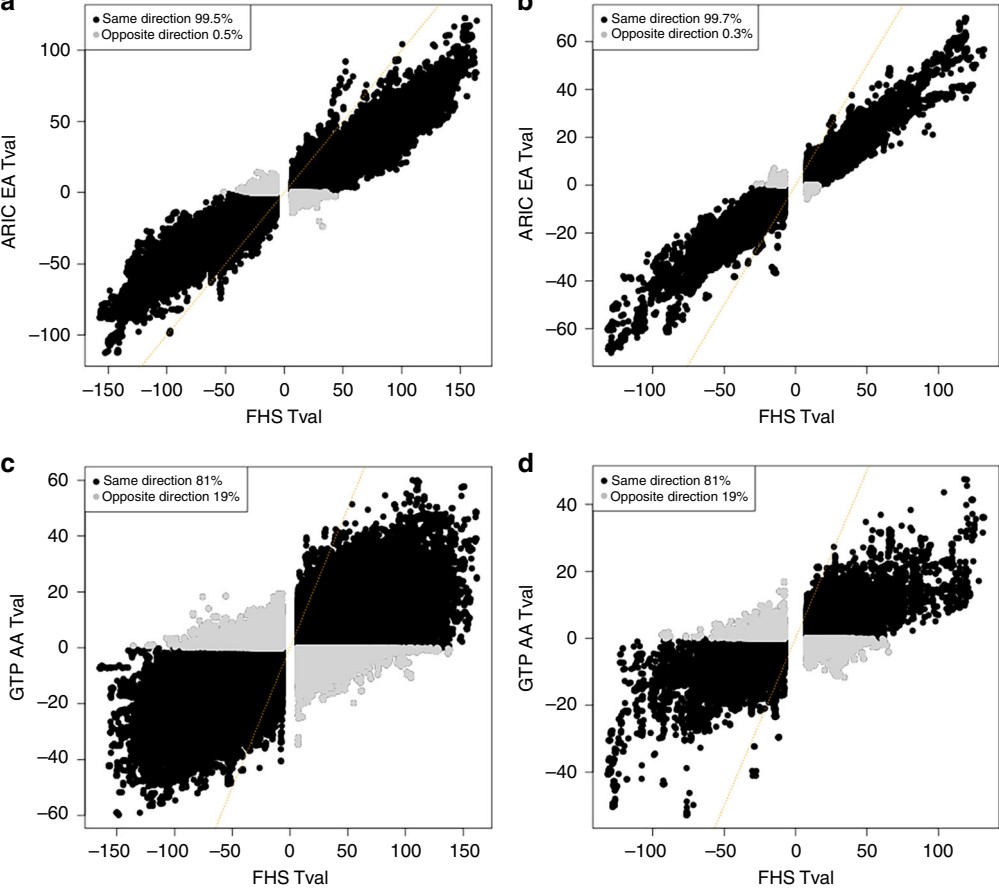

**Fig. 2** Plot estimates of *cis*- and *trans*-meQTLs in independent studies. **a** T-values of *cis*-meQTLs identified in FHS vs ARIC EA; **b** T-values of *trans*-meQTLs identified in FHS vs. ARIC EA; **c** T-values of *cis*-meQTLs identified in FHS vs GTP AA; **d** T-values of *trans*-meQTLs identified in FHS vs GTP AA

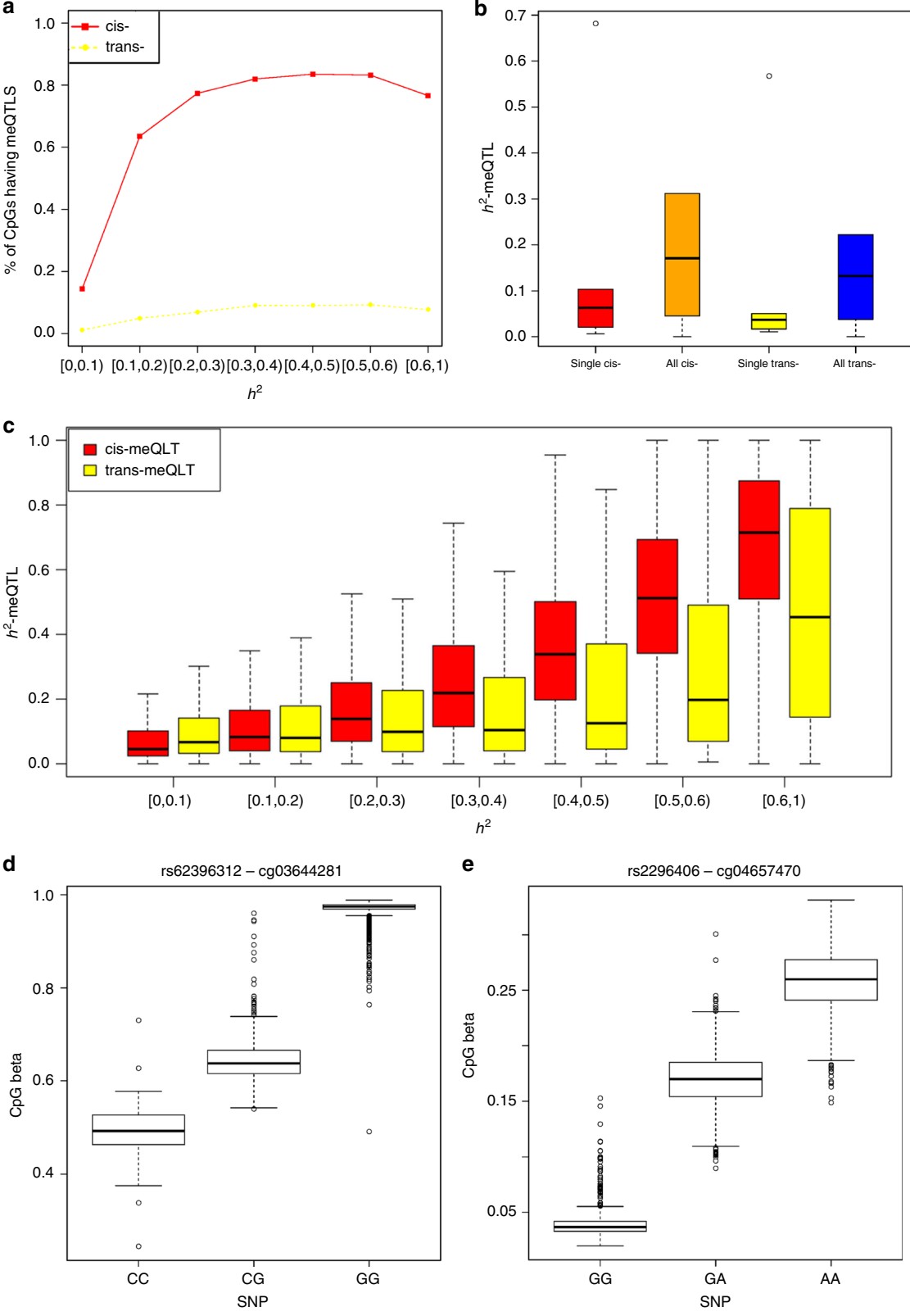

**Fig. 3** Characteristics of *cis*- and *trans*-meQTLs. **a** Proportion of CpGs having *cis*- and *trans*-meQTL SNPs at different $h^2_{CpG}$ levels; **b** Boxplot summary of $h^2_{meQTL}$ estimated by the peak *cis*-meQTL SNP, of all *cis*-meQTL SNPs, the peak *trans*-meQTL SNP and of all *trans*-meQTL SNPs. For each CpG, we chose one *cis*-meQTL SNP with the lowest *P*-value for the CpG as the peak *cis*-meQTL SNP, and one *trans*-meQTL SNP with the lowest *P*-value for the CpG as the peak *trans*-meQTL SNP. **c** Boxplot summary of $h^2_{meQTL}$ at different $h^2_{CpG}$ levels; **d** Boxplot of the *cis*-meQTL rs62396312–cg03644281; **e** Boxplot of the *trans*-meQTL rs2296406–cg04657470. Boxplots were drawn by the boxplot function in the *R* library. The boxes indicate the interquartile range (IQR) of values between the 75% (Q3) and 25% (Q1). The centre lines indicate the median value. The bars below and above each box indicate the data in Q1-1.5 x IQR and Q3 + 1.5 x IQR, respectively. For **d**, **e**, *y*-axis shows CpG methylation beta values, and *x*-axis shows SNP genotypes

Figure 4a shows that *cis*-meCpGs are enriched in promoter and enhancer regions (fold enrichment >1.2, $P < 1E-16$, hypergeometric test) and depleted in 5'UTR, TSS200, CpG islands, and high-CpG dense regions (fold enrichment <0.8, $P < 1E-16$, hypergeometric test). In contrast, *trans*-meCpGs are enriched for high-CpG dense regions (fold enrichment = 1.29, $P < 1E-16$, hypergeometric test) and depleted in 3'UTR and gene body regions (fold enrichment <0.8, $P < 1E-16$, hypergeometric test). Overlapping meQTL SNPs with Road-map project data[15] measured in primary cells and cell lines from peripheral blood and in many other tissues (see Methods) showed that the *cis*- and *trans*-meQTL SNPs are enriched for

active chromatin regions, such as transcription start site (TSS) active regions, transcription regions, enhancer regions, and ZNF genes and repeats regions, and highly depleted in heterochromatin and quiescent regions (FDR < 0.05 based on 1000 permutations; Fig. 4b, c and Supplementary Fig 6). The *cis*- and *trans*-meQTL SNPs are also enriched for eQTL SNPs[16,17] ($P < 1E-16$, hypergeometric test), protein QTL (pQTLs) SNPs[18,19] ($P < 1E-7$, hypergeometric test), and GWAS Catalog SNPs[20] ($P < 1E-16$, hypergeometric test). We found that 87% of *cis*-eQTL SNPs, 82% of *cis*-pQTL SNPs, and 59% of GWAS Catalog SNPs are also *cis*-meQTL variants (Supplementary Table 3).

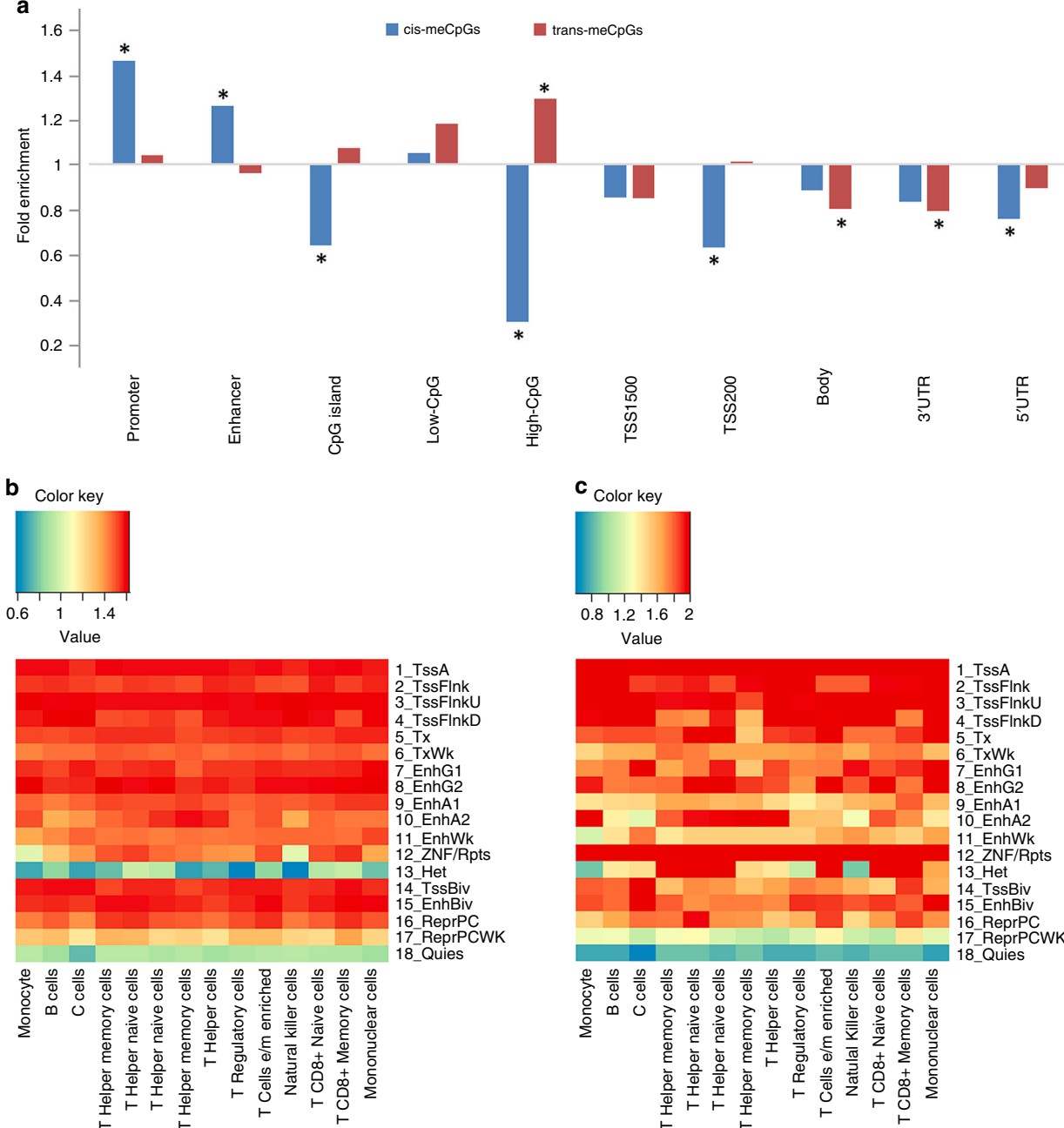

**Fig. 4** Genomic features enrichment analysis. **a** Enrichment of *cis*- and *trans*-meQTLs CpGs (meCpGs) in different genomic regions. Red indicated positively associated and green indicated negatively associated. * Indicate significant results with fold change > 1.2 or < 0.8, and $P < 0.05/10$. **b** Enrichment of *cis*-meQTL SNPs in different chromatin states in 14 blood cell lines, including monocyte, B cells, T cells, T helper memory cells, T helper naive cells, T helper naive cells, T helper memory cells, T helper cells, T regulatory cells, T cells e/m enriched, Natural killer cells, T CD8+ naive cells, T CD8+ memory Cells, and mononuclear cells; **c** Enrichment of *trans*-meQTL SNPs in different chromatin states

**Using *cis*-meQTLs to identify causal CpGs for CVD risk**. A large proportion (59%) of GWAS Catalog index SNPs were found to be *cis*-meQTL SNPs, indicating that a majority of phenotype-associated SNPs may contribute to disease pathways via effects on local DNAm. We further illustrated in two-sample MR analyses that *cis*-meQTL variants can be used as instrumental variables (IVs) to identify causal epigenetic mechanisms contributing to CVD and its metabolic risk factors (Fig. 5a). To derive IVs, we identified 159 thousand independent *cis*-meQTL SNPs pruned by LD $r^2 < 0.01$. By overlapping independent *cis*-meQTL SNPs with results from GWAS of CVD and its risk factors[21–25], we identified 14,910 CpGs, each of which has at least three independent *cis*-meQTL SNPs that were suitable instruments for MR to test for causal effects of DNAm on coronary heart disease (CHD)[21], myocardial infarction (MI)[21], type-II diabetes (T2D)[22], systolic (SBP) and diastolic blood pressure (DBP)[25], and 9921 CpGs suitable to test causality of DNAm on lipids traits including high-density lipoprotein (HDL) cholesterol, low-density lipoprotein (LDL) cholesterol, total cholesterol (TC), triglycerides (TG)[24], and body mass index (BMI)[23].

CpGs located within 2 Mb and highly correlated with each other ($r^2 \geq 0.5$) shared nearly 100% of their *cis*-meQTLs and were considered likely to affect the outcome trait through the same biological mechanism. We, therefore, pruned these CpGs and reported results for the CpG with the lowest *P*-value in MR testing as the index or putative causal CpG for a given window. In contrast, CpGs located close to each other (< 2 Mb) but with moderate correlations ($0 < r^2 < 0.5$) might be more likely to contribute through different regulatory mechanisms and were considered as independent epigenetic sites, even though they shared a partial set of *cis*-meQTL variants. We used multivariable MR methods to simultaneously estimate the independent causal effect of such CpGs on the outcome.

After correction for multiple testing ($P < 0.05/14{,}910$ or $0.05/9921$), MR analysis identified 92 putatively causal CpGs for CVD and its risk factors, including 12 putatively causal CpGs for CHD and MI, four for BMI, 33 for lipids traits, 37 for BP, and eight for T2D. Table 1 shows CHD and MI results and Supplementary Data 3 shows the full list (Supplementary Table 4 shows the multivariable MR results). Among the 12 putatively casual CpGs for CHD and MI, we found five CpGs that were positively associated with CHD/MI risk (e.g., cg24267699 in *ABO*, OR = 2.89, $P_{MR} = 1.34E{-}6$), and seven CpGs inversely associated with CHD/MI risk (e.g., cg09803321 in *NT5C2*, OR = 0.28, $P_{MR} = 3.75E{-}15$). Figure 5b lists 30 CpGs that were causal for more than two CVD phenotypes. A striking example is cg16306978 (*APOB*), which tested causal (positive direction of effect) for CHD (OR = 2.46, $P_{MR} = 2.1E{-}9$), LDL ($\beta = 1.08$, $P_{MR} = 2.9E{-}24$), TC ($\beta = 0.98$, $P_{MR} = 3.3E{-}22$), and TG ($\beta = 0.31$, $P_{MR} = 2.6E{-}6$). Another example was cg00908766 (*CESLR2*), which tested causal (positive direction of effect) for CHD (OR = 2.18, $P_{MR} = 4.3E{-}8$) and MI (OR = 2.01, $P_{MR} = 2.3E{-}6$) and inversely causal for HDL ($\beta = -0.29$, $P_{MR} = 2.0E{-}13$).

There were 2951 *cis*-meQTL SNPs that also were reported to be *cis*-eQTL variants in GTEx for multiple tissues[26] (fold enrichment = 1.2, $P < 1E{-}16$, hypergeometry test). Pathways analysis by FUMA (Functional mapping and annotation of GWAS)[27] revealed that the *cis*-meQTLs for the 92 putatively causal CpGs for CVD traits were over-represented with genes involved in sterol metabolic process, regulation of lipoprotein lipase activity, and glycine, serine, and threonine metabolism (Supplementary Data 4).

**Identify mRNAs involved in the causal pathways for CVD**. We performed comprehensive association analyses of 415 thousand CpGs with expression of ~18,000 mRNAs measured in 3,684 individuals (Chen Y. et al Unpublished) and found that *cis* associated CpG-mRNA pairs were replicable in independent external studies, whereas the *trans* pairs were not. Therefore, in this study, we only focused on mRNAs that were associated in *cis*

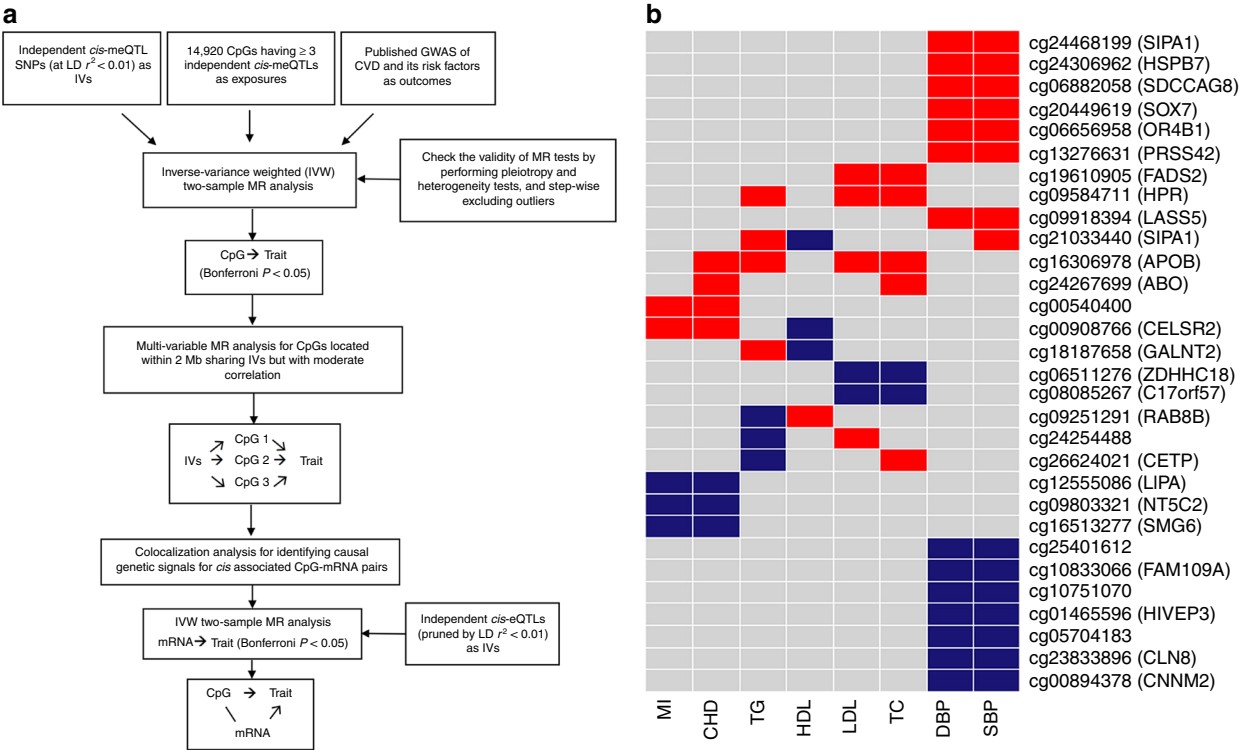

**Fig. 5** Mendelian randomization analysis using *cis*-meQTLs as causal anchors. **a** Analysis work flow; **b** Heatmap of 30 CpGs causal for more than two CVD risk factors. The red color shows positive directional effects and the blue color shows negative directional effects

**Table 1 Mendelian randomization results of coronary heart disease and myocardial infarction**

| CpG | Phenotype | Chr | Gene | Number of independent cis-meQTLs | IVW MR test OR | IVW MR test 95% CI | IVW MR test P-value | IVW MR test Bonferroni-corrected P-value | Heterogeneity test P-value | Pleiotropy test P-value |
|---|---|---|---|---|---|---|---|---|---|---|
| cg09803321 | CHD /MI | 10 | NT5C2 | 3 | 0.28 | 0.20–0.38 | 3.75E−15 | 5.59E−11 | 0.98 | 0.83 |
| cg12555086 | CHD /MI | 10 | LIPA | 4 | 0.42 | 0.32–0.54 | 1.41E−11 | 2.10E−07 | 0.62 | 0.37 |
| cg18534077 | CHD | 10 | AS3MT | 8 | 0.14 | 0.08–0.26 | 6.02E−11 | 8.98E−07 | 0.26 | 0.50 |
| cg02493740 | CHD | 2 | VAMP5 | 7 | 0.33 | 0.23–0.48 | 1.91E−09 | 2.85E−05 | 0.53 | 0.81 |
| cg16306978 | CHD | 2 | APOB | 3 | 2.46 | 1.83–3.30 | 2.09E−09 | 3.12E−05 | 0.73 | 0.48 |
| cg00908766 | CHD /MI | 1 | CELSR2 | 5 | 2.18 | 1.66–2.87 | 4.34E−08 | 6.47E−04 | 0.12 | 0.55 |
| cg00540400 | CHD /MI | 15 | | 3 | 3.00 | 2.03–4.45 | 5.99E−08 | 8.93E−04 | 0.38 | 0.57 |
| cg16513277 | CHD /MI | 17 | SMG6 | 3 | 0.06 | 0.02–0.17 | 1.99E−07 | 2.97E−03 | 0.62 | 0.43 |
| cg21433558 | CHD | 17 | CNTNAP1 | 5 | 2.69 | 1.82–3.98 | 6.73E−07 | 1.00E−02 | 0.70 | 0.36 |
| cg14037218 | CHD | 1 | ADAMTSL4 | 3 | 0.14 | 0.07–0.31 | 1.20E−06 | 1.79E−02 | 0.81 | 0.53 |
| cg24267699 | CHD | 9 | ABO | 4 | 2.89 | 1.88–4.44 | 1.34E−06 | 2.00E−02 | 0.61 | 0.47 |
| cg21692620 | MI | 17 | CNTNAP1 | 4 | 0.26 | 0.15–0.45 | 1.41E−06 | 2.10E−02 | 0.57 | 0.57 |

For CpGs that tested causal for both MI and CHD, only the MR results for CHD are shown in this table. The full MR results are shown in Supplementary Data 3

Bonferroni-corrected P-value is corrected for the number of CpGs having ≥3 independent cis-meQTLs (N = 14,910)

Independent cis-meQTLs were defined using LD $r^2 < 0.01$

with the 92 CpGs that tested positive in MR analyses. Among the 92 CpGs, 26 were associated in *cis* with 29 mRNAs in 3,684 individuals, including 35 CpG-mRNA pairs. Colocalization analysis revealed eight *cis*-associated CpG-mRNA pairs (including eight CpGs and eight mRNAs) for which DNAm and gene expression changes were driven by the same causal genetic variant, i.e., the causal *cis*-meQTL variant was also the causal *cis*-eQTL variant with a probability of >80%.

For these eight mRNAs, we used *cis*-eQTL SNPs as IVs in MR analysis to test if the expression changes are causal for CVD traits. At $P < 0.05/8$, we identified five genes whose expression was causal for CVD traits in whole blood using FHS-eQTL variants[16] (Supplementary Data 5), and five genes reflecting multiple tissues using GTEx eQTL variants[26] (Supplementary Table 5). Our results show for example that cg12555086, located in the *LIPA* gene body region, is associated in *cis* with expression of *LIPA* ($\beta = -4.4$, and $P = 1E-277$). A *cis*-meQTL variant associated with cg12555086 colocalized with a *cis*-eQTL variant associated with *LIPA* expression at a probability >0.99. *LIPA* expression tested causal for both CHD (OR = 0.42, and $P_{MR} = 1.4E-11$) and MI (OR = 0.36, and $P_{MR} = 6.1E-12$) in whole blood (Fig. 6a, b and Supplementary Fig 7a, b). The expression levels of *LIPA* were also tested casual for CHD and MI in many other tissues including adrenal gland, aorta, and liver. Another example is cg06882058 located in the gene body region of *SDCCAG8*; it is associated in *cis* with expression of *SDCCAG8* ($\beta = 1.72$, and $P = 1E-22$). A *cis*-meQTL variant associated with cg06882058 colocalized with a *cis*-eQTL variant associated with expression of *SDCCAG8* (colocalization probability > 0.99). *SDCCAG8* expression tested causal for both SBP ($\beta = 13.46$, and $P_{MR} = 7.2E-8$,) and DBP ($\beta = 10.40$, and $P_{MR} = 1.6E-7$) in whole blood (Fig. 6c, d and Supplementary Fig 7c, d). The expression levels of *SDCCAG8* also tested casual for SBP and DBP in many other tissues including tibial artery, heart atrial appendage, aorta, and brain. These results highlight many putatively causal pathways for CVD traits involving both DNAm and gene expression changes. Further experimental validation is needed to definitively prove causality.

**trans-meQTLs target local gene expression**. The molecular mechanism underlying *trans*-meQTLs is unknown. In a previous study, Shi et al reported a SNP (rs12933229) that was associated with five CpGs residing on different chromosomes[12]. In another study, Bonder et al explored *trans*-meQTL variants among 6111 GWAS Catalog SNPs and proposed that some genetic variants that affect the activity of a transcription factor in *cis* were associated in *trans* with DNAm changes at its binding sites[11]. In the present study, we identified 630 thousand *trans*-meQTL variants (6 million SNP-CpG pairs) genome-wide (Supplementary Fig 8) and demonstrated that *trans*-meQTLs are replicable in independent external studies.

Among the 630 thousand *trans*-meQTL variants that we report, 547 thousand (89%) also were *cis*-meQTL variants, and 178 thousand (28%) were *cis*-eQTL variants (enrichment test at $P < 1E-16$, hypergeometry test). The multiple roles of meQTLs lead us to hypothesize that SNPs with *trans*-acting effects on remote CpGs may do so via effects on nearby genes. We detected 22 *trans*-meQTL hotspots (denoted as H1 to H22), defined on the basis of an index SNP associated with at least 30 *trans*-meCpGs (Supplementary Data 6 and Fig. 7a). The 22 *trans*-meQTL hotspots targeted a total of 1701 *trans*-meCpGs. A total of 28% (875/3077) of the *trans*-meQTL hotspots variants were also *cis*-eQTLs associated with expression levels of 74 nearby genes (*cis*-eGenes). There were 146 *cis*-eGene-*trans*-meCpG associated pairs (at multiple testing corrected $P < 0.05$, including 19 *cis*-eGenes linked to 130 *trans*-meCpGs) in 3684 FHS participants. A hypergeometric test suggested that among all CpGs ($n = 901$) associated with *cis*-eGenes, there was enrichment for *trans*-meCpGs ($n = 130$, $P < 1E-16$, hypergeometry test). Gene ontology enrichment analysis showed that the 74 eGenes were enriched for transcription regulatory genes (24 genes, H2: *TCF7L1*; H13: *ZNF200, ZNF75A*; H14: *INO80E*; H19: *ZNF177, ZNF266, ZNF561*; H20: *ZNF333*; H21; *ZNF100, ZNF208, ZNF429, ZNF492, ZNF493, ZNF738*; H22: *HKR1, ZFP30, ZNF260, ZNF540, ZNF566, ZNF573, ZNF585B, ZNF607, ZNF781, ZEN793*) and DNA binding genes (22 genes, H3: *SP140L*; H13: *ZNF75A*; H17: *E2F4*; H19: *ZNF177, ZNF266*; H20: *ZNF333*; H21; *ZNF208, ZNF429, ZNF493; ZNF738*; H22: *HKR1, ZNF260, ZNF781, ZNF607, ZNF540, ZNF566, ZNF573*; Supplementary Table 6). One possible explanation for this observation is that the *trans*-meQTL SNPs may affect transcription regulatory genes in *cis*, and these transcription regulatory genes interact with *trans*-meCpGs to regulate their target genes (Fig. 7b). Further functional experiments are needed to prove this hypothesis.

## Discussion

Using a large single-site community-based cohort, we identified over 4.7 million *cis*- and 630 thousand *trans*-meQTL variants, roughly 3.5 times (*cis*-) and 10 times (*trans*-) more than the previous studies[10,11,13]. Independent external replication revealed that a large proportion of *cis*- and *trans*-meQTLs are replicable and the majority of our meQTL-CpG pairs showed directional concordance in independent external cohorts of participants of EA (99%) and AA (81%)

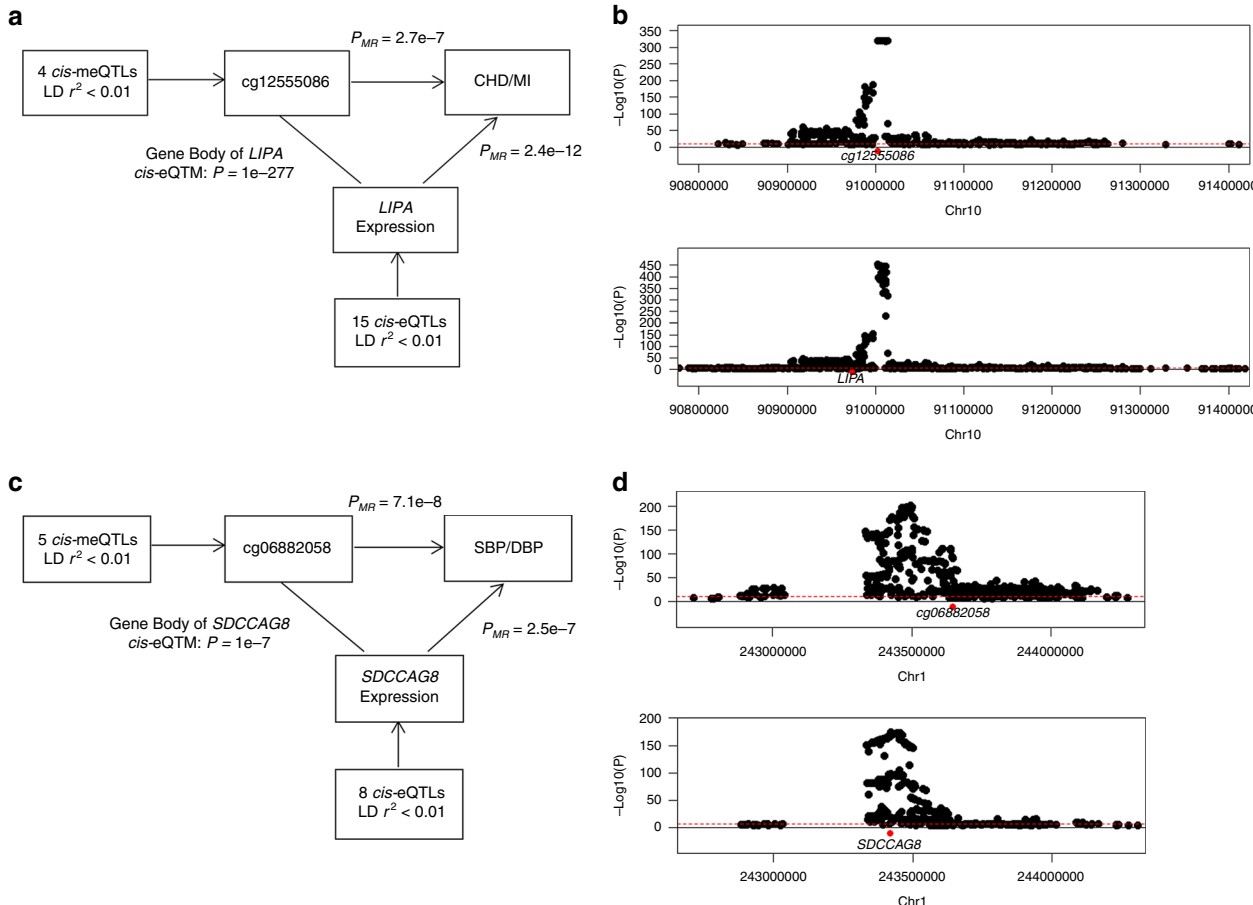

**Fig. 6** Mendelian randomization examples. **a** MR example of cg12555086 in relation to CHD and MI; **b** colocalization of a casual *cis*-meQTL and a casual *cis*-eQTL on cg12555086 and *LIPA* expression; **c** MR example of cg06882058 in relation to SBP and DBP; **d** colocalization of a casual *cis*-meQTL and a casual *cis*-eQTL on cg06882058 and *SDCCAG8* expression

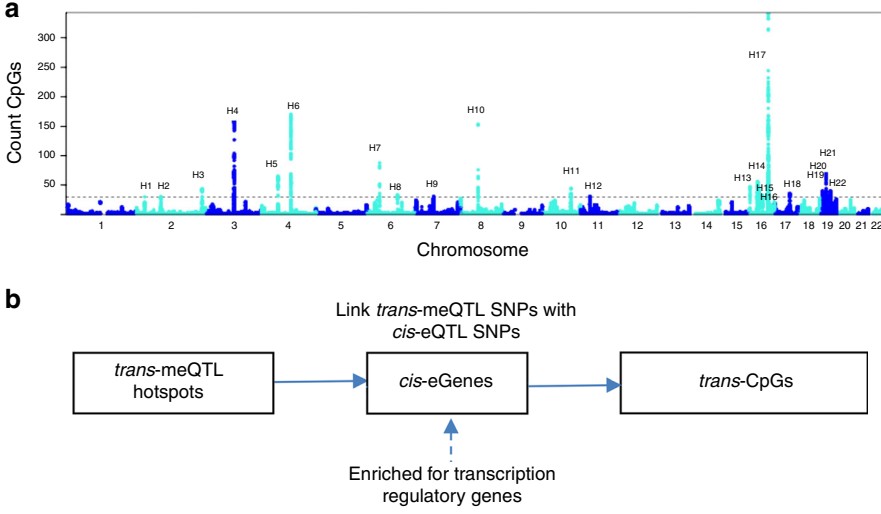

**Fig. 7** Overview and characteristics of *trans*-meQTLs. **a** *trans*-meQTL hotspots; **b** Linking *trans*-meQTLs with *cis*-eGenes

ancestry. The comprehensive meQTL resources provided by our study reveal a new richness of detail regarding genetic effects on DNAm patterns and the potential causal relation of epigenetic influences on various disease phenotypes. In this way, we help bridge a GWAS gap regarding disease-associated SNPs.

Our study is a well-powered multi-generational study. Thus, we were able to report accurate heritability estimates of DNAm in peripheral blood. We found that the average $h^2_{CpG}$ of all 415 thousand CpGs is 0.09, which is comparable to the values reported in a twins study (0.18–0.19)[28] and two family-based studies (0.13–0.14)[29,30]. Mapping of meQTLs suggested that

CpGs with higher $h^2_{CpG}$ were more likely to have *cis*- and *trans*-meQTL SNPs. The proportion of interindividual variation in methylation of CpGs explained by *cis*- and *trans*-meQTL SNPs increased proportionally with $h^2_{CpG}$.

The meQTLs identified in the FHS show excellent replicability in independent studies. As shown in Fig. 2, meQTLs from the FHS show 99% concordance in the ARIC EA cohort and 81% in the GTP AA cohort. Because the sample sizes in ARIC ($n =$ 963) and GTP ($n = 384$) were much smaller than in the FHS ($n = 4170$), the trend for T values of meQTLs was skewed and reflects the obvious benefits of using a large cohort in terms of greater statistical power to identify novel meQTLs. It is also the primary reason that our study identified many more *cis*- and *trans*-meQTLs than previous studies. Our study replicated the majority of meQTL SNP-CpG pairs identified in prior studies[10,11,13] (Supplementary Fig 5, 87% of *cis*- reported by Bonder et al.[11], 91% of *cis*- and 89% *trans*-pairs reported by Gaunt et al.[13], and 81% of *trans*-pairs reported by Lemire et al.[10]). Our study only replicated 36% of reported peak *cis*-meQTLs from Lemire et al. This low value might be because the peak *cis*-meQTLs in Lemire et al may not be as robust as others that could be largely replicated in our study. We acknowledge this as a limitation of our study that genotypes were from 1000 Genomes imputation data. Whole-genome sequencing methods for determining true genotypes rather than imputation will offer higher resolution and reduce the imputation uncertainty. Our study only replicated 10% of *trans*-meQTLs from Bonder et al, perhaps because they only focused on *trans*-meQTLs for GWAS Catalog SNPs and used a much lower *P*-value threshold ($P < 2.6E-7$) in their study versus $P < 1.5E-14$ in our study. The failure to replicate the remaining 10–20% *cis*- and *trans*-meQTLs identified in those previous studies might be due to different cohorts, genotype imputation accuracy, different statistical significance thresholds for meQTLs, or hidden confounding in the methylation data. Another limitation in our study is the limited coverage of the Illumina HumanMethylation450 platform, which may result in many missed meQTLs for CpGs that were not measured.

To demonstrate the utility of our meQTL resource, we explored several key CVD-related traits and performed MR to identify examples of CpGs that are putatively causal for CVD, which remains the leading cause of death worldwide[31]. Recently, several EWAS have reported CpGs associated with CVD risk factors[8,32–34], but the reported associations might not reflect causal effects owing to the inability to exclude possible alternative mechanisms, such as downstream effects of disease on methylation, or unexplained confounding factors. In fact, the vast majority of trait-associated CpGs from EWAS are more likely to reflect effects of the trait on DNA methylation rather than causal effects of methylation on disease[8,9,35]. A large database of meQTL variants enabled us to conduct systematic MR analyses using *cis*-meQTL variants as IVs to identify causal effects of DNAm on CVD traits. A similar recent study by Richardson et al[36] reported MR analysis using single *cis*-meQTL variants as IVs by focusing on 10 CpGs pre-selected based on EWAS and GWAS. Richardson et al acknowledged that the single IV MR method cannot separate causality from horizontal pleiotropy where genetic variants affect the exposure and outcome simultaneously[36]. In our study, because of the large number of meQTLs discovered, we were able to explore the direction of causality for more than 14 thousand candidate CpGs on CVD risk factors using multiple independent *cis*-meQTL variants ($\geq 3$) as IVs. In comparison with single-IV MR methods, the multiple-variant IV method increased the proportion of the variance in CVD traits explained versus a single variant and reduced the bias of horizontal pleiotropy that can affect single *cis*-meQTL MR analysis. Therefore, our approach has greater power to identify causal CpGs. Among all CpGs that tested causal for CVD traits, 24% have *cis*-meQTL variants that also are genome-wide significant ($P < 5E-8$) in GWAS of CVD traits (Table 1 and Supplementary Data 7). *cis*-meQTLs for the other causal CpGs were moderately associated with CVD and its risk factors in GWAS.

Some of the causal CpGs identified in this study reside in known CVD-related genes. For example, we identified cg12555086 for *LIPA* as being causal for CHD and MI. *LIPA* encodes lipase A, which is also known as cholesterol ester hydrolase. This enzyme functions in the lysosome to catalyze the hydrolysis of cholesteryl esters and triglycerides. Loss-of-function mutations in *LIPA* result in accelerated atherosclerosis[37]. Another example is six CpGs at the *ABO* locus that we found to be casual for CHD, MI, and total cholesterol levels (Supplementary Data 3). The *ABO* locus has been reported to be associated with CVD traits in previous studies[21,38]. ABO blood type has previously been linked to CVD risk in the FHS[39]. These published studies support our findings. In addition, our study provides evidence that changing expression levels for these genes may contribute to CVD risk. Some CpGs that tested positive by MR have not previously been reported to play a causal role in CVD. For example, we identified cg06882058 in *SDCCAG8* as causal for both SBP and DBP. CpG cg06882058 is associated in *cis* with expression of *SDCCAG8*, which we found to be causal for both SBP and DBP (Supplementary Data 5 and Supplementary Table 5). Previous studies found that *SDCCAG8* causes nephronophthisis type 10, characterized by retinal and renal degeneration, mild intellectual disability, obesity, hypogonadism, and recurrent respiratory infections in humans[40,41]. *Sdccag8* knockout mice develop late-onset nephronophthisis and severely increased BP[42]. This evidence leads us to hypothesize that dysregulation of cg06882058 may affect expression of *SDCCAG8* and thereby cause hypertension and contribute to CVD risk. Further experimental validation is necessary to prove this hypothesis.

There is considerable merit for using meQTLs along with other molecular QTLs, such as eQTLs to reveal much broader and more complex gene networks underlying genetic variant-disease associations. We show in the colocalization analysis that CpGs and their *cis*-associated gene expression are driven by the same causal variants. This suggests the presence of "vertical causal pathways" linking genetic variants, DNAm, and gene expression to human diseases.

Previous studies have revealed that DNAm patterns are highly tissue specific[43]. An important limitation of our study is that we may not detect many causal CpGs for CVD in whole blood-derived DNA when their contributions to disease is due to altered methylation in other tissues that are relevant to CVD. By overlapping meQTL SNPs with Epigenome Roadmap Project data (Fig. 4 and Supplementary Fig 6), our results show that the *cis*- and *trans*-meQTLs are enriched in transcription active regions and depleted in heterochromatin and quiescent regions in multiple tissues. It seems plausible that the putatively causal pathways derived by CpGs identified from whole blood are shared across tissues. MR testing utilizing GTEx eQTL variants[26] further confirms that many mRNAs lying on the CpG-derived pathways for CVD in whole blood were also causal for CVD traits in multiple tissues, including heart, liver, adipose, and others (Supplementary Data 5 and Supplementary Table 5).

Our study did not find causal CpGs among the top reported results of recently published EWAS of BP[34], lipids[33], and BMI[8,32]. As discussed above, CpGs reported by EWAS are more likely to reflect downstream effects of the trait on DNAm rather than causal effects of DNAm on disease[8,9,35]. Published EWAS may also have limited power to observe associations of causal CpGs with traits, perhaps because causal effects of CpGs on disease are smaller than the effects of disease on methylation of CpGs. One of

the most promising results of our study is for cg16306978 in *APOB*, an LDL particle ligand, which tested causal for CHD, LDL, TC, and TG (Supplementary Data 3). *APOB* protein has been linked to long-term CVD risk[18]. We found 12 of the 92 causal CpGs to be associated with smoking[6], and six CpGs were associated with alcohol consumption[7] (at $P < 0.05/92$; Supplementary Table 7). Our results may bridge a knowledge gap by explaining how environmental influences can alter epigenetic patterns that in turn affect diseases.

Our *trans*-meQTL results revealed an abundance of *trans*-meQTL hotspots and illustrated their putative activity on proximal nuclear binding genes and transcriptional regulatory genes. Our results are consistent with findings reported by Bonder et al.[11] and by Lemire et al.[10]. However, the *trans*-meQTLs reported by Bonder et al were limited to *trans*-meQTL variants that also are GWAS Catalog SNPs[11]. Lemire's study identified fewer than 2000 *trans*-meQTL SNP-CpG pairs due to a smaller sample size[36]. In contrast, we systematically mapped 22 *trans*-meQTLs hotspots that targeted more than 30 CpGs on different chromosomes. We replicated Shi et al.'s *trans*-meQTL hotspot finding (H14 in our results)[12]. Another promising example in our study, H17 (with a peak SNP rs7203742 located in an intron of *CTCF* on chromosome 16, Fig. 7a) is *cis*-acting on 17 genes, including several transcription regulatory genes, *E2F4*, *NUTF2*, and *NFATC3*, targeting 343 CpGs across the genome. rs7203742 was also identified by Lemire et al. as a *trans*-meQTL targeting 14 CpGs[10]. We speculate that these transcriptional regulatory genes may either coincide with their *trans*-associated CpGs to regulate downstream gene expression or involve direct regulation of DNAm of *trans*-associated CpGs. Further functional experiments are needed to explore molecular mechanisms underlying such *trans* phenomena. In our previous study, we identified 13 *trans*-eQTL hotspots that affected hundreds of genes[44]. Our previously reported *trans*-eQTLs hotspots did not overlap with the *trans*-meQTL hotspots identified in the present study, and the *cis*-eGenes affected by *trans*-eQTL hotspots did not show enrichment of transcriptional regulatory genes. Instead, the *trans*-eQTL hotspots were enriched for platelet SNPs and platelet eQTL variants. This finding indicates that *trans*-eQTLs are highly tissue specific, in contrast to the *trans*-meQTLs, which reflect remote control by transcriptional regulatory genes.

## Methods

**Study populations**. The discovery study used 4170 EA participants from the FHS, a community-based study of cardiovascular disease and its risk factors[45]. In 1971, the Offspring Generation cohort was recruited, consisting of the immediate descendants (and their spouses) of the Generation 1 cohort[46]. From 2002 to 2005, the Third Generation cohort was recruited, consisting of immediate descendants of the Offspring Generation cohort participants[47]. In this study, eligible participants included participants from the Offspring Generation cohort who attended their eighth examination cycle (Exam 8, 2005–2008, $N = 2648$), and Third Generation cohort participants who attended their second examination cycle (Exam 2, 2008–2011, $N = 1522$). This study was approved under Boston University Medical Center protocol H-27984. Written informed consent was obtained from each participant.

**DNA methylation profiling and data normalization**. DNA samples were extracted from whole blood buffy coat samples using the Gentra Puregene DNA extraction kit (Qiagen, Venlo, Netherland) and subsequently underwent bisulfite conversion using the EZ DNA methylation kit (Zymo Research, Irvine, CA). Samples underwent whole-genome amplification, fragmentation, array hybridization, and single-base pair extension. DNA methylation levels were measured using the Illumina Infinium Human Methylation450 BeadChip (450 K). FHS offspring cohort samples were run in two laboratory batches at the Johns Hopkins Center for Inherited Disease Research (lab batch #1) and University of Minnesota Biomedical Genomics Center (lab batch #2). DNA methylation arrays of the FHS Third Generation cohort samples (lab batch #3) were run by Illumina (San Diego, CA, USA).

For each lab batch, DNAm β were normalized using the DASEN methodology implemented in the wateRmelon R package, and the output β values for each CpG

were used in downstream analysis[48]. For sample quality control, we excluded (1) samples with missing methylation values (detection $P > 0.01$) at >1% CpGs, (2) samples with poor matching between the 65 single nucleotide polymorphisms (SNPs) on the Illumina 450 K array and the GWAS array, and 3) samples containing outliers at the multi-dimensional scaling plot. A total of 4170 samples passed final QC, including 2648 Offspring Generation cohort samples and 1522 Third Generation cohort samples. For QC at the probe level, we excluded probes with missing methylation values (detection $P > 0.01$) at >20% samples, probes previously identified to map to multiple locations[49] on sex chromosomes, and probes with an underlying SNP (minor allele frequency [MAF] > 5% in EA 1000 Genomes Project data) at the CpG site or within 10 bp of the single-base extension[50]. A total of 415,318 CpGs were retained for further analysis.

**Genotyping and genotype imputation**. Genotyping was performed using the Affymetrix 500 K mapping array and the Affymetrix 50K gene-focused MIP array. Quality control was conducted as described previously[51]. Genotypes were imputed from the 1000 Genomes Project panel phase 2 consisting of approximately 36.3 million SNPs using MACH / Minimac software[52]. SNPs with MAF > 0.01 and imputation quality ratio >0.3 were retained, resulting in approximately 8.5 million SNPs that were used for further meQTL mapping.

**meQTL mapping**. Because of the computational burden of running linear mixed effects (LME) models for 8.5 million SNPs × 415K CpGs, we adapted a two-step analysis strategy. Step one consisted of pre-adjusting the normalized DNAm β values for age, sex, top 50 methylation principle components (PC), predicted blood cell fraction[53], and pedigree by LME implemented in the R package Pedigreemm[54], from which the resulting residuals were retained. Then, linear regression was implemented using a Java script to test the associations between DNAm residuals and the imputed SNP dosage. The top 50 methylation PCs were chosen to maximize the internal replication rate across discovery-validation split samples (50–50%) in FHS. SNP-DNAm pairs residing within 1 Mb (*cis*) and those residing more than 1 Mb apart (*trans*) were identified separately. We used liberal $P$-value thresholds to pre-filter the meQTLs at $P < 1E-6$ for *cis* and $P < 1E-10$ for *trans*. Step two consisted of using the LME model implemented in the *lmekin()* function from the kinship2 R package[55] to re-calculate the associations between SNPs and DNAm for the pre-selected SNP-DNAm pairs by modeling the SNP as the independent variable, DNAm as the outcome, and adjusting for the same covariates as in step 1 as well as familial relatedness. The coefficients of co-ancestry were 0.5 between parent and offspring, 0.5 between siblings, and 0 between all other individuals. Bonferroni correction was used to maintain an experiment-wide type I error rate of 0.05 for $2.5 \times 10^9$ *cis* SNP-DNAm pairs and (415 K CpGs × 8.5 M SNPs – $2.5 \times 10^9$ *cis*-pairs) *trans* pairs, respectively, based on the CpGs and SNPs from the February 2009 assembly of the human genome (hg19, GRCh37 Genome Reference Consortium Human Reference 37). The corresponding significance thresholds are $P < 2E-11$ $[0.05/(2.5 \times 10^9)]$ for *cis*- and $P < 1.5E-14$ $[0.05/3.5 \times 10^{12}]$ for *trans*-meQTLs.

The estimates of $h^2_{cis-meQTLs}$ (i.e. the proportion of the total variance in a CpG explained by all of its *cis*-meQTL variants) and $h^2_{trans-meQTLs}$ (i.e. the proportion of the total variance in a CpG explained by all of its *trans*-meQTL variants) were calculated using 779 independent individuals by selecting one individual from each family in the FHS. To reduce the computational burden, we only used *cis*- and *trans*-meQTL SNPs that were genotyped. $h^2_{cis-meQTLs}$ and $h^2_{trans-meQTLs}$ were calculated using the *GREML* function in the GCTA package[56], adjusting for age, sex, top 50 methylation PCs and predicted blood cell fraction.

**Determination of the genomic control factor**. The genomic control factor (λ) was defined previously[57]. In our study, due to extensive storage requirements (i.e., storage of at least half of the raw results) to compute λ, we computed λ based on associations of 415 K CpGs with a random subsample of 100,000 SNPs selected from the 1000 Genomes Project panel with MAF > 0.01 and imputation quality ratio >0.3 and also within the HapMap2 SNP set.

**mRNA expression data**. Whole blood samples (2.5 ml) were collected in PAXgene™ tubes (PreAnalytiX, Hombrechtikon, Switzerland). mRNA expression was profiled using the Affymetrix Human Exon 1.0 ST GeneChip platform. Raw gene expression data were first normalized using the RMA (robust multi-array average) from Affymetrix Power Tools (APT, thermofisher.com/us/en/home/life-science/microarray-analysis/affymetrix.html#1_2) with quantile normalization. Then output expression values of 17,318 genes were extracted by APT based on NetAffx annotation version 31[58]. The gene expression values were adjusted for a set of technical covariates, e.g. chip batch, by fitting LME models as described previously[59]. mRNA expression data were available for 5626 individuals from the FHS Offspring cohort (Exam 8, $N = 2446$) and Third Generation cohort (Exam 2, $N = 3180$).

**Estimating the heritability of DNAm**. The narrow-sense heritability, estimated for the methylation level of individual CpGs (denoted as $h^2_{CpG}$), was the proportion of total phenotypic genetic variance ($\sigma^2_{CpG}$) due to the additive polygenic genetic

variance ($\sigma_A^2$): $h_{CpG}^2 = \sigma_A^2/\sigma_{CpG}^2$. $\sigma_{CpG}^2 = \sigma_A^2 + \sigma_h^2 + \sigma_e^2$, where $\sigma_h^2$ is the household variance and $\sigma_e^2$ is the environmental variance. LME regression models (using the *lmekin()* function of the *kinship2* R Package[55]) were used to estimate $h_{CpG}^2$ by modeling methylation values from the familial relatedness matrix with age, sex, top 50 methylation PCs, cell type fractions, and a household matrix as covariates. The coefficients of co-ancestry in the familial relatedness matrix are 0.5 between parent and offspring, 0.5 between siblings, and 0 between all other individuals. The household effect defined as the proportion of variances in a CpG was attributed to the individuals sharing household. The coefficient of shared household is 1 between individuals in the same family, and 0 between all other individuals.

**MR tests for the relations of DNAm and CVD.** Two-sample Mendelian randomization (MR) was used to identify putatively causal CpGs for CVD and its risk factors using a multi-step strategy. Estimated associations and effect sizes between SNPs and traits were based on the latest published meta-analysis GWAS of coronary heart disease (CHD)[21]; myocardial infarction (MI)[21]; type-2 diabetes (T2D)[22]; body mass index (BMI)[23]; lipids traits including high-density lipoprotein (HDL) cholesterol, low-density lipoprotein (LDL) cholesterol, total cholesterol (TC), and triglycerides (TG)[24]; systolic blood pressure (SBP); and diastolic blood pressure (DBP)[25]. Instrumental variables (IVs) for each CpG site were composed of independent *cis*-meQTLs pruned at LD $r^2 < 0.01$, retaining only the *cis*-meQTL variant with the lowest SNP-CpG *P*-value in each LD block. The LD proxies were defined using 1000 genomes European samples[60]. Inverse variance weighted (IVW) MR tests were performed on CpGs with at least three independent *cis*-meQTL variants, which is the minimum number of IVs needed to perform multi-instrument MR. For CHD, MI, SBP, DBP, and T2D GWAS results based on 1000 Genome Project reference panels, MR tests were performed on 14,910 CpGs. For lipid traits and BMI, with GWAS results based on the Metabochip genotyping platform, MR tests were performed on 9,921 CpGs, with at least 3 independent *cis*-meQTLs on the Metabochip. A large proportion of meQTLs were not measured on the Metabochip genotyping platform, which limits MR analyses for BMI and lipid traits. When the manuscript was under preparation, a new GWAS for CHD was published that was not incorporated in the MR analysis[61].

To test the validity of IVW-MR results, we performed heterogeneity and MR-EGGER pleiotropy tests for all IVs[62]. We used a step-wise strategy to select valid IVs for MR. If either $P_{heter}$ or $P_{pleio}$ were less than 0.05, we excluded the top outlier IV. IVW-MR, heterogeneity, and pleiotropy tests were repeated using the remaining IVs. Finally, significant causal CpGs for a trait were required to meet the following restrictions: (1) have at least 3 independent SNPs, (2) Bonferroni-corrected $P_{MR} < 0.05$, corrected for 14,910 for CHD, MI, SBP, DBP, and T2D, or Bonferroni-corrected $P_{MR} < 0.05$, corrected for 9,921 for lipids traits and BMI, and 3) both $P_{heter}$ and $P_{pleio} > 0.05$.

Because the *cis*-meQTLs were identified using a 2 Mb window, it is possible that CpGs located within 2 Mb shared *cis*-meQTLs (IVs). CpGs located in close proximity may be highly correlated and involved in the similar biological pathways. The remaining CpGs, although partially influenced by same set of genetic variants, exhibited only moderate correlation ($r^2 < 0.5$), and should be considered as independent epigenetic loci, raising the question of "horizontal pleiotropy IVs" for traditional MR analysis. To overcome this problem, we further applied multivariable MR methods[63] to moderately correlated CpGs located within 2 Mb of each other, and tested against the same outcome to simultaneously estimate the causal effect of each CpG on the outcome. The multivariable MR method is analogous to the simultaneous assessment of several treatments in a factorial randomized trial, and was successfully applied to simultaneously estimate causal effects of different lipid fractions on CVD[63]. MR analyses were conducted using the MRbase package in R[64].

**Identification of associations of gene expression and DNAm.** Association tests of DNAm and gene expression were performed in 3684 FHS participants with available DNAm and mRNA data. DNAm β values were adjusted for age, sex, predicted blood cell fraction, top 2 PCs of DNAm, and 25 surrogate variables (SVs), with DNAm as fixed effects, and batch as random effects by fitting LME models. Residuals (DNAm_resid) were retained. The gene expression values (RMA, see "mRNA expression data" section) were adjusted for age, sex, predicted blood cell fraction, a set of technical covariates[59], the two top PCs and 25 SVs, with gene expression as fixed effects, and batch as random effects by LME, and residuals (mRNA_resid) were retained. Then, linear regression models were used to assess pair-wise associations between DNAm_resid and mRNA_resid. SVs were calculated using the SVA package in R[65]. The significant threshold is Bonferroni-corrected $P < 0.05$ ($0.05/[415 \text{ K CpGs} \times 18 \text{ K mRNAs}] = 6.2 \times 10^{-12}$). A *cis*-CpG-mRNA pair was defined as a CpG residing $\pm 1$ Mb of the TSS of the corresponding gene encoding the mRNA (*cis*-eQTM). The annotations of CpGs and transcripts were obtained from annotation files of the HumanMethylation450K BeadChip and the Affymetrix exon array S1.0 platforms.

**Colocalization analysis.** Colocalization analysis was performed on *cis*-CpG-mRNA pairs (see *Identification of associations of gene expression and DNAm*) for the 92 putatively causal CpGs for CVD traits identified by MR (see *Mendelian randomization test for relationships of CpGs and CVD phenotypes*). Colocalization

tests were performed on each *cis*-CpG-mRNA pair using corresponding *cis*-meQTL and *cis*-eQTL variants. A Bayesian colocalization method implemented in the *coloc* R package was used to test the probability of one distinct variant being causal for both the CpG and gene expression[66], by assigning each *cis*-SNP-CpG-mRNA pair to one of the five hypothesis. H0: there exist no causal variants for either the CpG or the mRNA; H1: there exists a causal variant for the CpG only; H2: there exists a causal variant for the mRNA only; H3: there exist two distinct causal variants, one for the CpG and one for the mRNA; or H4: there exists a single causal variant for both the CpG and the mRNA. The result of this procedure is five posterior probabilities (PP0, PP1, PP2, PP3 and PP4) for each hypothesis. In this study, the association results between SNPs and DNAm (at $P < 1E-6$), and between SNPs and gene expression (at $P < 1E-4$) within a 2 Mb region were used as input. Prior probabilities were set as suggested by previous studies[18,67] –the probability of a SNP being associated with trait 1 only (p1) was $2E-11$, the probability of a SNP being associated with trait 2 only (p2) was $1E-7$, the probability of a SNP being associated with both traits (p12) was p1 multiplied by 10%. The p12 = p1 × 10% indicates the probability of a causal SNP for DNAm also being the causal SNP for gene expression at 10%. Those parameters revealed almost the same results when p1 and p2 are both set to $1E-4$, and p12 is set to $1E-5$. A SNP was considered to be colocalized for CpG and mRNA if the posterior probability (PP4) was greater than or equal to 80%.

**MR tests for the relations of gene expression to CVD.** To evaluate if gene expression is causal for CVD phenotypes in whole blood, independent *cis*-eQTLs (pruned by LD $r^2 < 0.01$, $n \geq 3$) form FHS eQTL resource[16] were used as IVs and IVW-MR was performed to test if changes in gene expression levels were causal for CVD phenotypes. To carefully select IVs unconfounded by heterogeneity and pleiotropy, we used the same strategy as described in the Mendelian randomization test for the relationships between CpG and CVD phenotypes section.

In order to test if gene expression in other tissues is causal for CVD phenotypes, we used *cis*-eQTL variants identified from 44 tissues by GTEx[26] as IVs. IVW-MR tests were used in relation to gene expression when more than three independent *cis*-eQTL variants (LD $r^2 < 0.01$) were available. If there were fewer than three independent *cis*-eQTL variants for gene expression available, we used the top *cis*-eQTL variant as an IV for MR testing. The statistical significance threshold was $P_{MR} < 0.05/8$.

**Functional annotation of CpGs and meQTLs.** Mapping and annotation of CpGs on the HumanMethylation450K BeadChip has been described previously[49]. Genomic features of CpGs were annotated, including CpGs located in CpG Islands, low or high CpG regions, promoter, enhancer, gene body, 3 prime untranslated region (3'UTR), 5'UTR, 0–200 bases upstream of transcription start sites (TSS200), and TSS1500. Hypergeometric tests were used to evaluate if the identified *cis*- and *trans*-meQTL showed enrichment for CpGs annotated with those genomic features. The significance threshold was defined by a fold change of >1.2 or <0.8 and a Bonferroni-corrected $P < 0.05/10$.

Epigenome Roadmap Project data[15] were used to determine whether the detected *cis*- and *trans*-meQTL SNPs were enriched for functional regions in the genome. We used data from primary cell lines of peripheral blood, including E029 monocytes, E032 B-cells, E034 T-cells, E037 T-helper memory cells, E038 T-helper naive cells, E039 T-helper naive cells, E040 T-helper memory cells, E043 T-helper cells, E044 T-regulatory cells, E045 primary T-cells effector/memory enriched, E046 natural killer cells, E047 T-CD8 + naive cells, E048 T-CD8+ memory cells, and E062 mononuclear cells, and from other CVD relevant tissues including E063 adipose nuclei, E065 aorta, E066 liver, E067 brain angular gyrus, E068 brain anterior caudate, E069 brain cingulate gyrus, E071 brain hippocampus middle, E072 brain inferior temporal lobe,E073 brain dorsolateral prefrontal cortex, E074 brain substantia nigra, E087 pancreatic islets, E095 left ventricle, E096 lung, E098 pancreas, E100 psoas muscle, E104 right atrium, E105 right ventricle, and E108 skeletal muscle. Eighteen chromatin states included active transcription start site (TssA), flanking transcription start site (TssFlnk), upstream flanking transcription start site (TssFlnkU), downstream flanking transcription start site (TssFlnkD), strong transcription (Tx), weak transcription (TxWk), genic enhancer (EnhG1 and EnhG2), active enhancer (EnhA1 and EnhA2), weak enhancer (EnhWk), zinc finger genes and repeats (ZNF_Rpts), heterochromatin (Het), bivalent/poised transcription start site (TssBiv), bivalent enhancer (EnhBiv), repressed polycomb (RepPC), weak repressed polycomb (RepPCWk), and quiescent (Quies).

To test whether the detected *cis*- and *trans*-meQTLs SNPs were enriched for SNPs residing in genomic regions annotated with chromatin states ("chromatin states SNPs"), we used a permutation-based strategy by randomly selecting equal numbers of MAF-matched SNPs 1000 times. Fold change was calculated as the ratio of the overlap between the tested-SNPs (i.e., *cis*- or *trans*-meQTL) and chromatin states SNPs to the overlap between the permutation-SNPs and chromatin states SNPs. The pools of candidate SNPs were from 1000-genomes imputed SNPs with MAF > 0.01 and imputation quality ratio >0.3 as described above. To match the distribution of MAFs of the permutation-SNP set with the tested-SNP set, we categorized MAF into four categories: MAF of (0.01, 0.05), (0.05, 0.1), (0.1, 0.2), and (0.2, 0.5). For each MAF category, we kept the proportion of SNPs in the permutation SNP set equal to the proportion of SNPs in the tested

SNP set. For *cis*-meQTLs, the proportions of SNPs in the four MAF categories were 18, 14, 21, and 47%, respectively. For *trans*-meQTLs, the proportions of SNPs in the four MAF categories were 12, 12, 21, and 55%, respectively. The statistical significance threshold was permutation-based FDR < 0.05 from 1000 permutation.

**Pathway analysis**. To investigate possible pathways underling the associations between meQTLs and CVD traits, we used FUMA (Functional mapping and annotation of GWAS)[27] on the *cis*-meQTLs for the putatively causal CpGs identified by MR analysis. FUMA included all loci with *cis*-meQTLs at LD > 0.8 based on the 1000 Genomes references panel as input. Hypergeometric tests on genes from those loci were used to investigate over representations of genes from multiple pathways. To improve focus in this study, we only use results of KEGG[68] and Gene Ontology—biological process (GO-BP) terms[69]. The SNP-to-Gene mapping was used on associations between SNPs and genes from positional mapping in Grch37/hg19, eQTLs in GTEx[26], and chromatin interaction mapping in Hi-C databases[70]. The significant threshold for the pathway analysis used a corrected P < 0.05/tests pathways in FUMA.

**Replication of *cis*- and *trans*-meQTLs in independent studies**. For the significant *cis*- and *trans*-meQTLs identified in FHS, we attempted replication of the SNP-CpG pairs in 963 ARIC EA participants and 384 GTP AA cohort participants. Due to lower numbers of samples in these cohorts, we excluded SNPs with MAF < 0.05 in ARIC and GTP.

ARIC Study is a prospective cohort conducted in four US communities to investigate atherosclerosis and clinical atherosclerotic diseases[71]. A total of 15,792 men and women aged 45 to 64 years (baseline) were recruited in 1987 and 1989 (visit 1) from four communities: Forsyth County, North Carolina; Jackson, Mississippi; suburbs of Minneapolis, Minnesota; and Washington County, Maryland. The ARIC study protocol was approved by the institutional review board of each participating university. Four subsequent follow-up exams were carried out in 1990–1992 (visit 2), 1993–1995 (visit 3), 1996–1998 (visit 4), and 2011–2013 (visit 5). DNA methylation was measured from visit 2 or visit3. Genomic DNA was extracted from peripheral whole blood samples using the Gentra Puregene Blood Kit (Qiagen; Valencia, CA, USA). Bisulfite-conversion of DNA samples was performed using the EZ-96 DNA Methylation Kit (Deep Well Format) (Zymo Research; Irvine, CA, USA), and then measured for methylation status using the Illumina HumanMethylation 450K beadarray (Illumina, Inc., San Diego, CA, USA). Probe intensities were extracted using Illumina GenomeStudio software (version 2011.1, Methylation module 1.9.0). Poor-quality samples with pass rate less than 95% were excluded. Samples were further excluded based on gender mismatch, SNP discordance with previous genotyping, and outliers in PC analysis. At the target level, poor-quality CpG sites missing in ≥5% samples were excluded. Beta values, representative of the methylation score for each CpG, were normalized using the Beta MIxture Quantile dilation (BMIQ) method[72]. Blood cell types were imputed using the Houseman method[73]. Genotyping was performed with Affymetrix SNP array 6.0. Imputation of missing genotypes was performed using IMPUTE2[74] using 1000G Phase 1 version 3 reference. We first fit LME models, where the BMIQ normalized beta values were the response variables, the covariates age, sex, visit number, blood cell counts, and the top 50 methylation PCs were fitted via fixed effects, and the batch factors chip ID, chip position, plate number were fitted as random effects. Residuals extracted from the fitted model were tested for associations with SNPs using linear regression models. All analysis were carried out in R.

GTP is a population-based prospective study to assess trauma exposure and stress-related outcomes in an urban, predominantly AA population[75]. Participants were recruited prospectively from the waiting rooms of primary care and obstetrics-gynecology clinics of Grady Memorial Hospital in Atlanta, GA. Since its inception in 2005, over 5000 participants have been interviewed for the study; 384 AA participants had both genotype data and whole blood DNA methylation measurements. Genotyping and DNA methylation profiling of GTP samples are described in the previous study[76]. Missing data points were defined by (1) a detection p-value greater than 0.001, or (2) a combined signal less than 25% of the total median signal and less than both the median unmethylated and median methylated signal. Individual samples were removed if they were outliers using a hierarchical clustering analysis, or had (1) a mean total signal less than half of the median of the overall mean signal or 2000 arbitrary units and (2) a missingness rate above 5%. Similarly, CpG probes were removed if the missingness rate was above 10% or the probes overlapped with SNPs (for Infinium I probes, SNPs at the site of single-base extension; for Infinium II probes, SNPs at CpG site; for both probes, SNPs located within 10 bp from site of single-base extension) with MAF > 0.05 in 1000 Genomes Project 20110521 release for African population (AFR)[50]. To remove the effect of outliers and ensure normality, DNAm β values were first inverse-normal transformed before analysis[77]. Associations between 437,229 CpGs and 9,892,561 SNPs (imputed allele dosage) were then estimated using linear regression adjusting age, gender and top 20 PCs of DNAm. The number of DNAm PCs was determined to achieve the highest power for *cis*-meQTLs mapped by at least one SNP ($P < 10^{-6}$, *cis*-meQTL was defined as CpG-SNP distance <1 Mb). All meQTL analyses were performed using the R package MatrixEQTL[78]. Associations between *cis*- and *trans*- SNP-DNAm pairs were identified and reported separately. The Institutional Review Boards of Emory University School of Medicine and

Grady Memorial Hospital approved the study protocol for the Grady Trauma Project.

We calculated the ratio of meQTLs replicated in ARIC and GTP at P < 0.05, P < 0.001, and at Bonferroni-corrected P < 0.05, corrected for 26.8 M tests for *cis* and 2 M tests for *trans*. We also compared the T-values of SNP-CpG pairs identified in FHS with T-values in ARIC and GTP.

**Data resources for eQTLs and pQTLs**. To examine the overlap of meQTLs with eQTLs and pQTLs, we used published resources of whole blood eQTLs identified in the FHS[16] and in a prior meta-analysis of European cohorts[17]. Plasma pQTLs were previously identified in the FHS[18] and KORA[19].

**Reporting summary**. Further information on research design is available in the Nature Research Reporting Summary linked to this article.

## Data availability

The complete set of DNAm data and mRNA expression data for FHS participants have been deposited in and are available from dbGaP under the study accession phs000724.v7.p11 (DNAm data) (https://www.ncbi.nlm.nih.gov/projects/gap/cgi-bin/study.cgi?study_id=phs000724.v7.p11) and phs000363.v17.p11 (mRNA expression data) (https://www.ncbi.nlm.nih.gov/projects/gap/cgi-bin/study.cgi?study_id=phs000363.v17.p11). The meQTL resources developed for this study are freely accessible via the NCBI Molecular QTL Browser (https://preview.ncbi.nlm.nih.gov/gap/eqtl/studies/) and via the NCBI ftp site (https://ftp.ncbi.nlm.nih.gov/eqtl/original_submissions/FHS_meQTLs/).

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

## Acknowledgements

The Framingham Heart Study is funded by National Institutes of Health contract N01-HC-25195 and HHSN268201500001I. The laboratory work for this investigation

was funded by the Division of Intramural Research, National Heart, Lung, and Blood Institute, National Institutes of Health. The analytical component of this project was funded by the Division of Intramural Research, National Heart, Lung, and Blood Institute, and the Center for Information Technology, National Institutes of Health, Bethesda, MD. M.M.M. is supported by a NHLBI K99HL136875. The Atherosclerosis Risk in Communities study has been funded in whole or in part with Federal funds from the National Heart, Lung, and Blood Institute, National Institutes of Health, Department of Health and Human Services (contract numbers HHSN268201700001I, HHSN268201700002I, HHSN268201700003I, HHSN268201700004I, and HHSN268201700005I). The authors thank the staff and participants of the ARIC study for their important contributions. Funding was also supported by 5RC2HL102419 and R01NS087541. Dr. Ci Song is supported by the international postdoc fellowship award from Swedish Research Council (2016-00598). The views expressed in this manuscript are those of the authors and do not necessarily represent the views of the National Heart, Lung, and Blood Institute; the National Institutes of Health; or the U.S. Department of Health and Human Services.

## Author contributions

D.L., T.H., and L.L. designed, directed, and supervised the project. T. H., L.L and D.L. drafted the papermanuscript. T.H., J.R., C.S., F.P. and Y.G. conducted the analyses. All authors participated in revising and editing the papermanuscripts. All authors have read and approved the final version of the papermanuscript.

## Additional information

**Competing interests:** The authors declare no competing interests.

