## [Peer Review File · Nature Communications]

Reviewers' Comments:

Reviewer #1:

Remarks to the Author:

Huan and co-workers analyzed genome-wide associations of genetic variants with whole blood DNA 55 methylation of 415K CpG sites in 4,170 FHS participants resulting 394 thousand independent cis- and 21 thousand independent trans- loci ($r^2 < 0.2$) that largely replicated in the ARIC study.

The authors claim that because of larger sample size they detected 3.5 times more cis- and 10 times more trans-meQTL SNPs than previous studies. Also 2/3 of the identified cis meQTL and >85% of the trans-meQTL are told to be novel. Thus an unprecedented meQTL database resource.

Then as proof-of concept, they show the utility of their meQTL database by linking cis-meQTL variants with GWAS results for cardiovascular disease (CVD) traits followed by Mendelian randomization analysis to infer causal relations between meQTL and these traits..

Major concerns

1. The authors claim to provide the largest mQTL database to date seems valid. This should serve as a good recourse for DNA methylation patterns in CVD in blood. However, methylation has repeatedly been shown to be highly tissue-specific; (<https://www.ncbi.nlm.nih.gov/pmc/articles/PMC3322498/> , <https://www.ncbi.nlm.nih.gov/pmc/articles/PMC5596466/> <https://www.ncbi.nlm.nih.gov/pmc/articles/PMC4053947/> , to reference a few studies on this topic). This fact is unfortunately completely ignored even in Discussion section where the last paragraph is describing study limitations. The importance of tissue-specific methylation is central for the general use of this repository and the identification causal genes for CVD traits. Most CVD traits are not believed to be governed by pathway/genes represented by white blood cells. Instead for plasma lipids, type 2 DM and atherosclerosis the main target tissues are liver, Beta cells/skeletal muscle and the arterial wall, respectively. However, many of the proposed genes inferred from the MR analysis are indeed already known target genes (e.g., LIPA, APOB) suggesting some blood cell methylation sites are shared across tissues.

Thus, it seems central for the general impact of this study (meriting publication in Nature Com.) to in depth analyze these earlier studies to dissect the extent of meQTLs find herein are shared with other tissues. It seems plausible that most sites for which the author find evidence of causal CDT trait genes should be shared with other tissues?

Depending how the results of this analysis emerge – the conclusion: that their results “highlight putatively causal pathways involved in the pathogenesis of human diseases and to identify promising therapeutic targets for disease prevention and treatment” may or may not be premature.

2. The authors also show in Figure 3A that CpGs with higher heritability are more likely to associate with cis and trans-meQTL SNPs as evidence for their disease relevance. It seems that a parallel assessment of H2 for the cis and trans- eQTLs in these regions is warranted to decipher if variation in methylation exceeds that of SNPs in the same region?

Minor concerns

1. Using linkage disequilibrium $r^2 < 0.2$ to define independent meQTLs – does this mean you considered all meQTLs in $LD > 0.8$ as dependent – if so a rather none-stringent threshold it seems?
2. If 87% cis eQTLs are also cis meQTL -what is the use of identifying cis meQTLs?

3. The 22 trans-meQTL hotspots seem interesting and novel.
4. The co-localization analysis is a nice complement to the study with results that seem compelling.
5. ABO and the novel gene SDCCAG8 (for MI and total chol) are interesting findings since both these genes appear to involve pathways where methylation patterns in blood are relevant (see also major concern 1).
6. Without being a statistician by training, it appears to me the data analyses are relevant and accurate. I assume, however, the manuscript may undergo a statistical review down the road.

Reviewer #2:

Remarks to the Author:

Profs Levy, Liang and co-workers have performed a high throughput DNA methylation QTL experiment, which I believe is the largest of its kind to date. Through extensive genotype imputation, the authors identify substantially more trans and cis meQTLs than previous studies and annotate these findings. Beyond discovering meQTLs they also attempt replication (although in smaller sample sizes). They perform Mendelian randomisation analyses for causal effects of CpGs on CVD traits confirming previously known CVD genes and suggesting probable new ones at CVD loci. The authors intend to make their data available to the community, which will form a valuable resource for the scientific community.

Overall the analyses appear to be robust and thorough. However, the manuscript would benefit from clarification, particularly in respect of presenting the results (e.g. Figures) and some further thought around the MR analyses.

Across the plots, the descriptions need to clearly state what the figures represent e.g. Figure 3D, "Boxplot of cis-meQTL SNP rs62396312 explained 60% of the variance in cg03644281" but I can't get this information from the plot. Where is the 60% represented? E.g. 2. Figure 4B, what does the yellow/orange etc colouring represent? Only mention green and red in the figure title (not good choice for red/green colour blind people). Might also wish to state cell types rather than the codes. E.G. 3 Figure 5B, what do the colours represent? E.G. 4 Figure 7 do the size of the dots mean anything? What's an mQTL, does it differ to an meQTL? What's a genotype address/CpG address and what is along the diagonal? E.G. 5 what SNPs do the black dots represent in Figure 6B,C, F, G?

There are many examples of lack of clarity of which I have highlighted just a few. Please review all figures closely.

For the MR analyses, I wonder why the authors didn't use MR-EGGER to test for pleiotropy? Figure 6G shows the effect estimate plots for the IVs (I think! Information not provided in figure legend) and on this plot neither the black or the red lines go through the origin suggesting pleiotropy. (I'm assuming either the black or the red line is the MR causal effect? It would be useful to have the $y=0$ and $x=0$ lines on these plots.) I'm surprised that the pleiotropy test did not highlight this pleiotropic effect. Please can the authors discuss? Did the authors consider outlier effects e.g. MR-PRESSO? The authors also mention multi-variable MR in the methods, are these results provided in the results?

The authors replicate most previous meQTLs, they could also discuss the reasons for lack of replication of previous reported meQTLs.

The genomic control is not a true representation of inflation in the test statistics as it uses just a small fraction of the data. Given replication of previous findings etc, I would be surprised if the authors have an issue with inflation but perhaps it would be more appropriate to increase the number of variants used for the GC calculation, e.g. extend to the genotyped set at least.

Minor points:

1. Line 120 pg 5, what is meant by "household effects"?
2. The authors, probably in the discussion should state that the replication data are statistically under-powered, this is almost certainly the reason for the skew in the Figure 2 plots.
3. What does BF represent? Suggest the authors define or preferentially use Bonferroni (which I guess this stands for?).
4. Peak cis-meQTL/peak trans-meQTL – what does this mean (fig 3)
5. Would be helpful if the authors restricted use of h2 to heritability rather than both heritability and variance explained, so that the distinction is clear.
6. Something has gone wrong with the figure label letters in Figure 6.
7. Page 23 line 522, what does "moderately correlated CpGs" mean? (Sounds subjective)
8. Page 24, colocalization description, it would be helpful for the reader for the description to include something about the hypotheses, H1-H4. Particularly the last sentence, it's the PP of H4 that provides information on colocalization. Likewise, line 557, what is multiple 10%. Please clarify the descriptions here.
9. Page 10/11 cis-eQTM is used, presumably a typo.

Reviewer #3:

Remarks to the Author:

Huan and colleagues leverage meQTLs to better understand cardiovascular disease. In their manuscript, they perform a large meQTL study and subsequently use a variety of methods and corroborating pieces of evidence to identify putatively causal genes. Overall, I think the manuscript provides a valuable meQTL resource for the field and combines meQTLs creatively with statistical genetics methods. My main comment is that there needs to be stronger emphasis in the manuscript text, and potentially a couple of additional analyses that really highlight the advantage of combining eQTLs, meQTLs, and GWAS over simply using eQTLs and GWAS alone. I think this is an important point to emphasize and support with analyses, because coupling eQTLs with GWAS is the standard in the variant-to-gene mapping field, but there is still a lot of room in the field for the inclusion of meQTLs and other types of QTLs. The key is that added value must be shown. I believe this manuscript has the potential to show that value with a couple of additional analyses.

1. Lines 136-137: interesting observation around most "trans" meQTLs being long range. How does this proportion of "true" vs "fake" trans-meQTLs compare with trans and cis eQTLs from the GTEx database?
2. Lines 152-160: Why did only ~1/3rd of meQTLs replicate? Were the ones that did not replicate in the two smaller studies due to differences in sample size, and thus an expected inability to identify small effect size meQTLs?
3. Lines 180-182: Aren't trans-meCpGs depleted near/in genes by definition? Is that really a biologically meaningful insight?
4. Line 192: It's a bit unclear if this is 59% of GWAS Catalog index SNPs, or 59% of SNPs in strong LD with GWAS Catalog index SNPs. Are you stating that 59% of the time, the top SNP is a meQTL?
5. Lines 228-251: I think it would be very helpful to see how eQTLs from other databases such as GTEx/Blueprint etc line up with the meQTLs you found to be associated with disease via MR.
6. Line 247: do you mean colocalized with SDCCAG8 eQTL?
7. Line 278: I would really like to see some kind of broad comparison between trans-meQTLs with trans-eQTLs to see if the conclusions are similar.
8. The discussion feels a little too specific on certain points. I would like to see a little bit more of conceptual ideas driving the end of the discussion section, especially around how coupling meQTLs and eQTLs can be advantageous for identifying disease-causal genes over simply using eQTLs and GWAS alone. The ending of the discussion feels abrupt with the limitations. I suggest folding the limitations throughout the discussion.

9. The figure labels for Figure 1B and 4A can be difficult to read when the labels overlap the bars. I suggest moving the labels above or below the bars
10. Figure 7A may be better suited for the supplement.
11. I suggest moving the Molecular QTL browser section into the discussion or methods.
12. I would like to see some pathway analysis on all of the putative causal genes identified in the manuscript beyond those that were just identified in the tran-meQTL results in the Best,

We thank you and the three Reviewers for constructive comments regarding our manuscript. We have revised our manuscript in accordance with your suggestions and reviewer's comments, specifically addressing the comments made by Reviewer 2 (Comment #6) regarding the validity of instrumental variables in MR analysis. Our detailed responses to Reviewers' questions are provided below. The reviewers' comments are highlighted in bold text.

In addition, we have ensured that our manuscript complies with the journal's policies and format requirements, and deposited our data in an enduring repository as described in the Data Availability section.

Detailed Responses to Reviewers' Comments

Reviewer 1:

1. The authors claim to provide the largest mQTL database to date seems valid. This should serve as a good recourse for DNA methylation patterns in CVD in blood. However, methylation has repeatedly been shown to be highly tissue-specific; (<https://www.ncbi.nlm.nih.gov/pmc/articles/PMC3322498/> , <https://www.ncbi.nlm.nih.gov/pmc/articles/PMC5596466/> , <https://www.ncbi.nlm.nih.gov/pmc/articles/PMC4053947/> , to reference a few studies on this topic). This fact is unfortunately completely ignored even in Discussion section where the last paragraph is describing study limitations. The importance of tissue-specific methylation is central for the general use of this repository and the identification causal genes for CVD traits. Most CVD traits are not believed to be governed by pathway/genes represented by white blood cells. Instead for plasma lipids, type 2 DM and atherosclerosis the main target tissues are liver, Beta cells/skeletal muscle and the arterial wall, respectively. However, many of the proposed genes inferred from the MR analysis are indeed already known target genes (e.g., *LIPA*, *APOB*) suggesting some blood cell methylation sites are shared across tissues.

Thus, it seems central for the general impact of this study (meriting publication in Nature Com.) to in depth analyze these earlier studies to dissect the extent of meQTLs find herein are shared with other tissues. It seems plausible that most sites for which the author find evidence of causal CDT trait genes should be shared with other tissues?

Depending how the results of this analysis emerge – the conclusion: that their results “highlight putatively causal pathways involved in the pathogenesis of human diseases and to identify promising therapeutic targets for disease prevention and treatment” may or may not be premature.

Reply: We thank the Reviewer for this critical comment. We agree with the Reviewer that DNA methylation patterns are tissue-specific. This is an important limitation of this study: meQTLs were identified from whole blood, which is not a primary tissue for studying many CVD traits. We now discuss this limitation in the revised manuscript on page 15 and as follows.

“Previous studies have revealed that DNAm patterns are highly tissue specific[1-3]. An important limitation of our study is that we may not detect many causal CpGs for CVD in whole blood-derived DNA when their contributions to disease is due to altered methylation in other tissues that are relevant to CVD.”

Previous eQTL studies have suggested that genetic effects on gene expression levels may be observed more than expected in different tissues [4, 5]. However, there is a paucity of large meQTL studies in non-blood tissues that prevents a fair comparison of meQTLs in different tissues. In the previous version of our manuscript, we showed that *cis*- and *trans*-meQTLs were enriched in active chromatin regions by overlapping meQTL SNPs with genomic regions classified into chromatin states by the Epigenome Roadmap Project [6] in 14 cell lines from peripheral blood (Fig 4B-C). To explore the activities of meQTLs in more CVD-relevant tissues, we further performed the enrichment tests in additional tissues using Roadmap data, including adipose, brain, heart, liver, lung, muscle, pancreatic islets, and pancreas. As shown in Supplementary Fig 6, the *cis*- and *trans*-meQTLs in our study show enrichment in active chromatin regions in those tissues, just as they do in whole blood. This finding suggests that the meQTLs identified in whole blood also act to impact DNA methylation in multiple tissues. Furthermore, we used GTEx eQTLs identified in 44 tissues [7] to explore the putatively causal relations of genes with CVD traits in different tissues by Mendelian randomization testing. In the revised manuscript, the new Results, Methods, and Discussion sections reflect these substantial changes as follows:

Methods section, page 25:

“Epigenome Roadmap Project data [6] were used to determine whether the detected cis- and trans-meQTL SNPs were enriched for functional regions in the genome. We used data from primary cell lines of peripheral blood, including E029 monocytes, E032 B-cells, E034 T-cells, E037 T-helper memory cells, E038 T-helper naive cells, E039 T-helper naive cells, E040 T-helper memory cells, E043 T-helper cells, E044 T-regulatory cells, E045 primary T-cells effector/memory enriched, E046 natural killer cells, E047 T-CD8+ naive cells, E048 T-CD8+ memory cells, and E062 mononuclear cells, and from other CVD relevant tissues including E063 adipose nuclei, E065 aorta, E066 liver, E067 brain angular gyrus, E068 brain anterior caudate, E069 brain cingulate gyrus, E071 brain hippocampus middle, E072 brain inferior temporal lobe, E073 brain dorsolateral prefrontal cortex, E074 brain substantia nigra, E087 pancreatic islets, E095 left ventricle, E096 lung, E098 pancreas, E100 psoas muscle, E104 right atrium, E105 right ventricle, and E108 skeletal muscle. Eighteen chromatin states included active transcription start site (TssA), flanking transcription start site (TssFlnk), upstream flanking transcription start site (TssFlnkU), downstream flanking transcription start site (TssFlnkD), strong transcription (Tx), weak transcription (TxWk), genic enhancer (EnhG1 and EnhG2), active enhancer (EnhA1 and EnhA2), weak enhancer (EnhWk), zinc finger genes and repeats (ZNF_Rpts), heterochromatin (Het), bivalent/poised transcription start site (TssBiv), bivalent enhancer (EnhBiv), repressed polycomb (RepPC), weak repressed polycomb (RepPCWk), and quiescent (Quies).”

Methods section: page 24-25:

“MR tests for the relations of gene expression to CVD phenotypes

...

In order to test if gene expression in other tissues is causal for CVD phenotypes, we used cis-eQTL variants identified from 44 tissues by GTEx [7] as IVs. IVW-MR tests were used in relation to gene expression when more than three independent cis-eQTL variants ($LD r^2 < 0.01$) were available. If there were fewer than 3 independent cis-eQTL variants for gene expression available, we used the top cis-eQTL SNP as an IV for MR testing. The statistical significance threshold was $P_{MR} < 0.05/8$.”

Results section: page 7:

“Overlapping meQTL SNPs with Roadmap project data [6] measured in primary cells and cell lines from peripheral blood and in many other tissues (see Methods) showed that the cis- and trans-meQTL SNPs are enriched for active chromatin regions, such as transcription start site (TSS) active regions,

transcription regions, enhancer regions, and ZNF genes and repeats regions, and highly depleted in heterochromatin and quiescent regions (FDR<0.05 based on 1000-times permutations; Fig 4B-C and Supplementary Fig 6).”

Results section: page 10:

“For these eight mRNAs, we used *cis*-eQTL SNPs as IVs in MR analysis to test if the expression changes are causal for CVD traits. At $P < 0.05/8$, we identified five genes whose expression was causal for CVD traits in whole blood using FHS-eQTL variants (Supplementary Data 9), and five genes reflecting multiple tissues using GTEx eQTL variants (Supplementary Data 10). Our results show for example that cg12555086, located in the LIPA gene body region, is associated in *cis* with the expression of LIPA ($\beta = -4.4$, and $P = 1e-277$). *cis*-meQTL variants associated with cg12555086 colocalized with *cis*-eQTL variants associated with LIPA expression at a probability > 0.99 . LIPA expression tested causal for both CHD (OR=0.42, and $P_{MR} = 1.4e-11$) and MI (OR=0.36, and $P_{MR} = 6.1e-12$) in whole blood (Fig 6A-D). The expression levels of LIPA also tested casual for CHD and MI in many other tissues including adrenal gland, aorta, and liver. Another example is cg06882058 located in the gene body region of SDCCAG8; it is associated in *cis* with the expression of SDCCAG8 ($\beta = 1.72$, and $P = 1e-22$) and colocalized with *cis*-eQTL variants associated with expression of SDCCAG8 (colocalization probability > 0.99). SDCCAG8 expression tested causal for both SBP ($\beta = 13.46$, and $P_{MR} = 7.2e-8$) and DBP ($\beta = 10.40$, and $P_{MR} = 1.6e-7$) in whole blood (Fig 6E-H). The expression levels of SDCCAG8 also tested casual for SBP and DBP in many other tissues including tibial artery, heart atrial appendage, aorta, and brain. These results highlight many putatively causal pathways for CVD traits involving both DNAm and gene expression changes. Further experimental validation is needed to definitively prove causality.”

Discussion section: page 15-16:

“Previous studies have revealed that DNAm patterns are highly tissue specific[1-3]. An important limitation of our study is that we may not detect many causal CpGs for CVD in whole blood-derived DNA when their contributions to disease is due to altered methylation in other tissues that are relevant to CVD. By overlapping meQTL SNPs with Epigenome Roadmap Project data (Fig 4 and Supplementary Fig 6), our results show that the *cis*- and *trans*-meQTLs are enriched in transcription active regions and depleted in heterochromatin and quiescent regions in multiple tissues. It seems plausible that the putatively causal pathways derived by CpGs identified from whole blood are shared across tissues. MR testing utilizing GTEx eQTL variants [7] further confirmed that many mRNAs lying on the CpG-derived pathways for CVD in whole blood were also causal for CVD traits in multiple tissues, including heart, liver, adipose, and others. (Supplementary Data 9-10).”

2. The authors also show in Figure 3A that CpGs with higher heritability are more likely to associate with *cis* and *trans*-meQTL SNPs as evidence for their disease relevance. It seems that a parallel assessment of H2 for the *cis* and *trans*- eQTLs in these regions is warranted to decipher if variation in methylation exceeds that of SNPs in the same region?

Reply: The reviewer is correct. Figure 3A shows that CpGs with higher heritability are more likely to be associated with *cis*- and *trans*-meQTL SNPs. This finding suggests that the major genetic variants linked to methylation could be from SNPs in those *cis* and *trans* regions. Figure 3C shows the positive correlation between h^2_{CpG} and $h^2_{cis- / trans-meQTLs}$. It also suggests that as the heritability of CpG methylation increases, so does the proportion of variance in CpG methylation explained by identified meQTLs.

3. Using linkage disequilibrium $r^2 < 0.2$ to define independent meQTLs – does this mean you considered all meQTLs in LD > 0.8 as dependent – if so a rather none-stringent threshold it seems?

Reply: We considered all meQTL variants at LD $r^2 > 0.2$ to be non-independent. Using this LD filter, we retained one meQTL SNP for a given CpG and the association was represented as on SNP-CpG pair.

4. If 87% of cis eQTLs are also cis meQTL -what is the use of identifying cis meQTLs?

Reply: A large proportion (87%) of *cis*-eQTL variants also are *cis*-meQTL variants, indicating that the same genetic variants might drive changes of both nearby DNA methylation and gene expression, or that the changes in gene expression might be caused by the changes of DNA methylation. However, the number of *cis*-meQTLs (4.7 million) is much larger than *cis*-eQTLs (1.15 million). Only 21% of *cis*-meQTLs are *cis*-eQTLs. This feature suggests that a large portion of genetic variants that affect methylation do not affect gene expression. The value of identification of *cis*-meQTLs beyond *cis*-eQTLs is to give an additional dimension with which to explore the involvement of DNA methylation in human diseases. However, future more comprehensive analysis should be performed to further address the relationships between DNA methylation and gene expression. Such efforts are beyond the scope of the current study.

5. The 22 trans-meQTL hotspots seem interesting and novel.

Reply: We thank the reviewer for this comment.

6. The co-localization analysis is a nice complement to the study with results that seem compelling.

Reply: We thank the reviewer for this comment.

7. ABO and the novel gene SDCCAG8 (for MI and total chol) are interesting findings since both these genes appear to involve pathways where methylation patterns in blood are relevant (see also major concern 1).

Reply: To address the tissue specificity issue, we used *cis*-eQTLs identified in multiple tissues from the GTEx database to infer the causal roles of genes for CVD traits across multiple tissues. The new results have been updated in the revised manuscript. We found that expression levels of *SDCCAG8* were putatively causal for systolic and diastolic blood pressure in many tissues such as adipose, tibial artery, heart atrial appendage, aorta, and brain. Please also see the reply to Reviewer 1, Comment #1.

8. Without being a statistician by training, it appears to me the data analyses are relevant and accurate. I assume, however, that the manuscript may undergo a statistical review down the road.

Reply: We thank the reviewer for this comment.

Reviewer 2:

1. Overall the analyses appear to be robust and thorough. However, the manuscript would benefit from clarification, particularly in respect of presenting the results (e.g. Figures) and some further thought around the MR analyses.

Across the plots, the descriptions need to clearly state what the figures represent e.g. Figure 3D, “Boxplot of cis-meQTL SNP rs62396312 explained 60% of the variance in cg03644281” but I can’t get this information from the plot. Where is the 60% represented?

Reply: We thank the reviewer for this comment. The boxplots do not show the proportion of the variance in CpGs explained by SNPs. We deleted the misleading descriptions from the figure legend. We modified the figure legend of Figure 3 as follows:

“Figure 3: Characteristics of cis- and trans-meQTLs. A) Proportion of CpGs having cis- and trans-meQTL SNPs at different h^2_{CpG} levels; B) Boxplot summary of h^2_{meQTL} estimated by the peak cis-meQTL SNP, of all cis-meQTL SNPs, the peak trans-meQTL SNP, and of all trans-meQTL SNPs. For each CpG, we chose one cis-meQTL SNP with the lowest P value for the CpG as the peak cis-meQTL SNP, and one trans-meQTL SNP with the lowest P value for the CpG as the peak trans-meQTL SNP. C) Boxplot summary of h^2_{meQTL} at different h^2_{CpG} levels; D) Boxplot of the cis-meQTL rs62396312 -- cg03644281; E) Boxplot of the trans-meQTL rs2296406 -- cg04657470. Boxplots were drawn by the boxplot function in the R library. The boxes indicate the interquartile range (IQR) of data between 75% (Q3) and 25% (Q1). The bars below and above each box indicate the data in $Q1-1.5 \times IQR$ and $Q3+1.5 \times IQR$ respectively. For Figures D and E, y-axis shows CpG methylation beta values, and x-axis shows SNP genotypes.”

2. Figure 4B, what does the yellow/orange etc colouring represent? Only mention green and red in the figure title (not good choice for red/green colour blind people). Might also wish to state cell types rather than the codes.

Reply: We revised Figure 4B-C by adding a color key to show the color scales representing the values of fold change from 0.6-0.8 (blue), 0.801-1.2 (yellow), and >1.2 (red). We also replaced labels in Figure 4B-C using cell types names.

3. Figure 5B, what do the colours represent?

Reply: We added the description in the figure legend of Figure 5B as follows:

“Figure 5: Mendelian randomization analysis using cis-meQTL variants as causal anchors. A) Analysis work flow; B) Heatmap of 30 CpGs causal for more than two CVD risk factors. The red color shows positive directional effects and the blue color shows negative directional effects.”

4. Figure 7 do the size of the dots mean anything? Whats an mQTL, does it differ to an meQTL? Whats a genotype address/CpG address and what is along the diagonal?

Reply: We revised the figure legends of Figure 7A (now listed as Supplementary Figure 7). “mQTL” is a typo. We corrected it as follows: “meQTL”. The figure legend is shown as follows:

“Supplementary Figure 7: 2-D regional plot of trans-meQTLs genome-widely. Each dot represents a trans-meQTL at $P < 1.5e-14$. The dot size is inversely proportional to the P values. The largest dot indicates $P < 1e-200$. The dots along the diagonal indicate that a SNP and CpG (for trans-meQTLs) reside in the same chromosome. X-axis: SNP locations in chromosome, y-axis: CpG locations in chromosome. The numbers along the x-axis and y-axis reflect chromosome number.”

5. What SNPs do the black dots represent in Figure 6B, C, F, G?

There are many examples of lack of clarity of which I have highlighted just a few. Please review all figures closely.

Reply: The figure legends of Figure 6B, C, F, G (now as Supplementary Figure 7A-D) were revised, as follows:

“Supplementary Figure 7: MR-Egger regression scatterplots for effects of exposure on outcome. A) cg12555086 on CHD; B) LIPA expression on CHD; C) cg06882058 on SBP; D) SDCCAG8 expression on SBP; Scatter plots demonstrate the relationships of SNP effects on exposure (i.e., CpG or gene expression) and on the outcome (i.e., CHD and SBP). Each black dot in the scatter plots represents a cis-meQTL / eQTL SNP, and the red dots show the independent SNPs pruned by LD $r^2 < 0.01$ as IVs, with standard error bars. The lines show the MR-Egger regression lines regressed from all cis-meQTLs / eQTLs (black) or independent cis-meQTLs / eQTLs (red). The estimated MR-Egger intercepts are non-significantly different from 0 at $P > 0.05$.”

In addition, we carefully checked the remaining figures (i.e., Figure 1 and 2, and Supplementary Figures) and figure legends in the manuscript and made sure to add all necessary information.

6. For the MR analyses, I wonder why the authors didn't use MR-EGGER to test for pleiotropy? Figure 6G shows the effect estimate plots for the IVs (I think! Information not provided in figure legend) and on this plot neither the black or the red lines go through the origin suggesting pleiotropy. (I'm assuming either the black or the red line is the MR causal effect? It would be useful to have the y=0 and x=0 lines on these plots.) I'm surprised that the pleiotropy test did not highlight this pleiotropic effect. Please can the authors discuss? Did the authors consider outlier effects e.g. MR-PRESSO? The authors also mention multi-variable MR in the methods, are these results provided in the results?

Reply: For the MR analysis, we reported the MR estimates from IVW-MR and used MR-EGGER to test for pleiotropic effects of IVs to ensure the validity of IVs. We used a step-wise strategy to exclude outliers as described in the Methods on page 22 and as provided below. Please also see reply to Review 2, Comment #5. For the top MR results reported in our manuscript (Supplementary Data 6), we further used the MR-PRESSO outlier test to confirm that there are no outliers among the IVs. We revised the manuscript and modified Figure 6 (now as Supplementary Figure 7) to indicate no pleiotropic effects of IVs in this study.

“To test the validity of IVW-MR results, we performed heterogeneity and MR-EGGER pleiotropy tests for all IVs[8]. We used a step-wise strategy to select valid IVs for MR [9]. If either P_{heter} or P_{pleio} were less than 0.05, we excluded the top outlier IV. IVW-MR, heterogeneity, and pleiotropy tests were repeated using the remaining IVs. Finally, significant causal CpGs for a trait were required to meet the following restrictions: 1) have at least 3 independent SNPs, 2) Bonferroni-corrected $P_{MR} < 0.05/14,910$ for CHD, MI, SBP, DBP, and T2D, or Bonferroni-corrected $P_{MR} < 0.05/9,921$ for lipids traits and BMI, and 3) both P_{heter} and $P_{pleio} > 0.05$.”

The multi-variable MR results are now provided in Supplementary Data 7.

7. The authors replicate most previous meQTLs, they could also discuss the reasons for lack of replication of previous reported meQTLs.

Reply: In the revised manuscript, we discuss the possible reasons for lack of replication of previous meQTLs, on page 13:

“Our study replicated the majority of meQTL SNP-CpG pairs identified in prior studies [11-13] (Supplementary Fig 5, 87% of cis- reported by Bonder et al, 91% of cis- and 89% trans-pairs reported by Gaunt et al, and 81% of trans-pairs reported by Lemire et al). Our study only replicated 36% of reported peak cis-meQTLs from Lemire et al. This low value might be because the peak cis-meQTLs in Lemire's

study may not be as robust as others that could be largely replicated in our study. We acknowledge this as a limitation of our study that genotypes were from 1000 Genomes imputation data. Whole genome sequencing methods for determining true genotypes rather than imputation will offer higher resolution and reduce the imputation uncertainty. Our study only replicated 10% of trans-meQTLs from Bonder et al, perhaps because they only focused on trans-meQTLs for GWAS Catalog SNPs and used a much lower P value threshold ($P < 2.6e-7$) in their study versus $P < 1.5e-14$ in our study. The failure to replicate the remaining 10-20% cis- and trans-meQTLs identified in those previous studies might be due to different cohorts, genotype imputation accuracy, different statistical significance thresholds for meQTLs, or some hidden factors in the methylation data. Another limitation of our study is the limited coverage of the Illumina HumanMethylation450 platform, which may result in many missed meQTLs for CpGs were not measured.”

8. The genomic control is not a true representation of inflation in the test statistics as it uses just a small fraction of the data. Given replication of previous findings etc, I would be surprised if the authors have an issue with inflation but perhaps it would be more appropriate to increase the number of variants used for the GC calculation, e.g. extend to the genotyped set at least.

Reply: The lambda GC value (lambda=0.93) in this study was computed based on 4.2×10^{11} tests, *i.e.*, 100K randomly chosen SNPs multiplied by 420K CpG sites. The number of tests was already much larger than is typical for a GWAS (*i.e.*, 10^7 tests per trait) and EWAS (*i.e.*, 10^6 tests) analyses. The primary difficulty for us to repeat the lambda GC calculation as the reviewer suggested, is the amount of computational memory this would need. Because we have to store practically everything needed for the ordering of the P-values. We believe that using 100K SNPs from the genome provides sufficient power to capture population stratification and other systematic bias such that the lambda GC value shown in the manuscript represents the distribution of P-values and the approximate median P-value of all SNPs very well.

9. Line 120 pg 5, what is meant by “household effects”?

Reply: We added the definition of household effects in Methods on page 21, as follows,

“The household effect was defined as the proportion of variance in a CpG that was attributed to individuals sharing a household. The coefficient of shared household is 1 between individuals in the same family, and 0 between all other individuals.”

10. The authors, probably in the discussion should state that the replication data are statistically under-powered, this is almost certainly the reason for the skew in the Figure 2 plots.

Reply: We agree with the reviewer. In the revised manuscript, we discuss this issue on page 13 as follows:

“The meQTLs identified in the FHS show excellent replicability in independent studies. As shown in Fig 2, meQTLs from the FHS show 99% concordance in the ARIC EA cohort and 81% in the GTP AA cohort. Because the sample sizes in ARIC (n=963) and GTP (n=384) were much smaller than in the FHS (n=4170) the trend for T values of meQTLs was skewed and reflects the obvious benefits of using a large cohort in terms of greater statistical power to identify novel meQTLs. It is also the primary reason that our study identified many more cis- and trans-meQTLs than previous studies.”

11. What does BF represent? Suggest the authors define or preferentially use Bonferroni (which I guess this stands for?).

Reply: We replaced “BF” with “Bonferroni” all through the manuscript.

12. Peak *cis*-meQTL/peak *trans*-meQTL – what does this mean (fig 3)

Reply: We added the definition of peak *cis*- and *trans*-meQTLs in the figure legend of Figure 3, as follows:

“For each CpG, we chose one cis-meQTL SNP with the lowest P value for the CpG as the peak cis-meQTL SNP, and one trans-meQTL SNP with the lowest p value for the CpG as the peak trans-meQTL SNP.”

13. Would be helpful if the authors restricted use of h^2 to heritability rather than both heritability and variance explained, so that the distinction is clear.

Reply: We thank reviewer for this suggestion. We restricted use of h^2 to indicate heritability rather than “variances explained” in the revised manuscript.

14. Something has gone wrong with the figure label letters in Figure 6.

Reply: We corrected the error.

15. Page 23 line 522, what does “moderately correlated CpGs” mean? (Sounds subjective)

Reply: In the Methods section on page 23, we defined “moderately correlated CpGs” as those with Pearson correlation $R^2 < 0.5$. Among CpGs located in close proximity (within 2 Mb) sharing *cis*-meQTLs, for those CpGs at $R^2 > 0.5$, we used one CpG as a representative CpG for this region. For those CpGs at $R^2 < 0.5$, we used multivariable MR methods and tested the same outcome to simultaneously estimate the causal effect of each CpG on the outcome.

16. Page 24, colocalization description, it would be helpful for the reader for the description to include something about the hypotheses, H1-H4. Particularly the last sentence, it’s the PP of H4 that provides information on colocalization. Likewise, line 557, what is multiple 10%. Please clarify the descriptions here.

Reply: We revised the methods sections on page 24, as follows:

“Colocalization tests were performed on each cis-CpG-mRNA pair using corresponding cis-meQTL and cis-eQTL variants. A Bayesian colocalization method implemented in the coloc R package was used to test the probability of one distinct variant being causal for both the CpG and gene expression [14], by assigning each cis-SNP-CpG-mRNA pair to one of the five hypothesis. H0: there exist no causal variants for either CpG or mRNA; H1: there exists a causal variant for CpG only; H2: there exists a causal variant for mRNA only; H3: there exist two distinct causal variants, one for CpG and one for mRNA; or H4: there exists a single causal variant for both the CpG and the mRNA. The result of this procedure is five posterior probabilities (PP0, PP1, PP2, PP3 and PP4) for each hypothesis. In this study, the association results between SNPs and DNAm (at $P < 1e-6$), and between SNPs and gene expression (at $P < 1e-4$) within a 2 Mb region were used as input. Prior probabilities were set as suggested by previous studies[15, 16] –the probability of a SNP being associated with trait 1 only ($p1$) was $2e-11$, the probability of a SNP being associated with trait 2 only ($p2$) was $1e-7$, the probability of a SNP being associated with both traits ($p12$) was $p1$ multiplied by 10%. The $p12 = p1 \times 10\%$ indicates the probability of a causal SNP for DNAm also being the causal SNP for gene expression at 10%. Those parameters revealed almost the same results when $p1$ and $p2$ are both set to $1e-4$, and $p12$ is set to $1e-5$. A SNP was considered to be colocalized for CpG and mRNA if the posterior probability (PP4) was greater than or equal to 80%.”

17. Page 10/11 cis-eQTM is used, presumably a typo.

Reply: *cis*-eQTM stands for *cis*- expression quantitative trait methylation. To avoid confusion from too many abbreviations, we did not use the term eQTM any more in the revised manuscript.

Reviewer 3:

1. My main comment is that there needs to be stronger emphasis in the manuscript text, and potentially a couple of additional analyses that really highlight the advantage of combining eQTLs, meQTLs, and GWAS over simply using eQTLs and GWAS alone. I think this is an important point to emphasize and support with analyses, because coupling eQTLs with GWAS is the standard in the variant-to-gene mapping field, but there is still a lot of room in the field for the inclusion of meQTLs and other types of QTLs. The key is that added value must be shown. I believe this manuscript has the potential to show that value with a couple of additional analyses.

Reply: We thank the reviewer for this valuable suggestion. In the revised manuscript, we have emphasized the value of combining meQTLs and eQTLs to infer causal pathways for human disease and traits. For example, in the Discussion on page 15, we include the following text:

“There is considerable merit for using meQTLs along with other molecular QTLs, such as eQTLs to reveal much broader and more complex gene networks underlying genetic variant – disease associations. We show in the colocalization analysis that CpGs and their cis-associated gene expression are driven by the same causal variants. This suggests the presence of “vertical causal pathways” linking genetic variants, DNAm, and gene expression to human diseases.”

We also performed additional analysis such as using GTEx *cis*-eQTLs for Mendelian randomization testing for genes related to CVD. Please see our reply to Reviewer #3, Comment #6.

2. Lines 136-137: interesting observation around most “trans” meQTLs being long range. How does this proportion of “true” vs “fake” trans-meQTLs compare with trans and cis eQTLs from the GTEx database?

Reply: In the GTEx data, *cis*-eQTLs are defined as variants residing within 2 Mb of the genes, and the *trans*-eQTLs are restricted to variants and genes lying on different chromosome [7]. Because GTEx did not provide eQTLs residing more than 2Mb from a gene in the same chromosome, we were unable to calculate the proportion of “true” vs “fake” *trans*-eQTLs as we did in using our own data. We speculate that this may be the reason for their definition of *trans*-eQTLs (different chromosome), because of the similar observation of a large proportion of “fake” *trans*-eQTLs or long-range *cis*-eQTLs in GTEx.

3. Lines 152-160: Why did only ~1/3rd of meQTLs replicate? Were the ones that did not replicate in the two smaller studies due to differences in sample size, and thus an expected inability to identify small effect size meQTLs?

Reply: In the revised manuscript, we discuss this issue on page 12 as follows:

“The meQTLs identified in the FHS showed excellent replicability in independent studies. As shown in Fig 2, meQTLs from the FHS showed 99% concordance in the ARIC EA cohort and 81% in the GTP AA cohort. Because the sample sizes in ARIC (n=963) and GTP (n=384) were much smaller than in the FHS (n=4170), the trend for T values of meQTLs was skewed and reflected the obvious benefits of using a large cohort in terms of greater statistical power to identify novel meQTLs. It is also the primary reason that our study identified many more cis- and trans-meQTLs than previous studies.”

4. Lines 180-182: Aren't trans-meCpGs depleted near/in genes by definition? Is that really a biologically meaningful insight?

Reply: We tested whether the *trans*-meCpGs were enriched or depleted in genomic functional regions (e.g. 3'UTR) of the genome. We did not test whether *trans*-meQTLs were depleted near / in genes by definition. The enrichment analysis suggested that *trans*-meCpGs were depleted in 3'UTR and gene body regions (fold enrichment <0.8, $P < 1e-16$) and enriched in the CpG-dense region (fold enrichment =1.29, $P < 1e-16$). In our *trans*-meQTL hotspots analysis (page 10-12), we found that *trans*-meQTL hotspots SNPs were also *cis*-eQTLs for transcription regulatory genes. Based on these results, we speculated that *trans*-meQTL SNPs may affect their nearby transcription regulatory genes, and those transcription regulatory genes in turn regulate *trans*-meCpGs. In this way, the alteration of *trans*-meCpG methylation may further affect nearby gene expression. If this hypothesis is true, we expect that *trans*-meCpGs will tend to reside in gene expression regulatory regions. Previous studies show evidence that CpGs residing in CpG-dense regions and CpG islands regulate nearby gene expression, rather than those residing in 3'UTR and gene body regions. The abnormal methylation of CpGs in gene body regions is more likely to be associated with diseases, such as cancer. Based on these features, we speculate that *trans*-meCpGs depleted in 3'UTR and gene body regions are more likely to be biologically meaningful.

5. Line 192: It's a bit unclear if this is 59% of GWAS Catalog index SNPs, or 59% of SNPs in strong LD with GWAS Catalog index SNPs. Are you stating that 59% of the time, the top SNP is a meQTL?

Reply: We showed that 59% of GWAS Catalog index SNPs overlapped with meQTL variants in our manuscript. We added the word “index” in the revised manuscript to avoid confusion. See page 8:

“A large proportion (59%) of GWAS catalog index SNPs were found to be cis-meQTL SNPs, indicating that a majority of phenotype-associated SNPs may contribute to disease pathways via effects on local DNAm.”

6. Lines 228-251: I think it would be very helpful to see how eQTLs from other databases such as GTEx/Blueprint etc line up with the meQTLs you found to be associated with disease via MR.

Reply: We thank the Reviewer for this suggestion. We checked the overlap of *cis*-meQTL variants for the putatively causal CpGs for CVD traits with GTEx eQTL variants. We also used the eQTLs from the GTEx database for MR to test mRNAs for causality for CVD traits in multiple tissues. The new results and methods are provided as follows:

In Methods on page 24-25:

“MR tests for the relations of gene expression to CVD phenotypes

...

In order to test if gene expression in other tissues is causal for CVD phenotypes, we used cis-eQTL variants identified from 44 tissues by GTEx [7] as IVs. IVW-MR tests were used in relation to gene expression when more than three independent cis-eQTLs ($LD r^2 < 0.01$) were available. If there were

fewer than three independent cis-eQTL variants for gene expression available, we used the top cis-eQTL variant as an IV for MR testing. The statistical significance threshold was $P_{MR} < 0.05/8$.

In Results on page 9:

“There were 2951 cis-meQTL SNPs that also were reported to be cis-eQTL variants in GTEx for multiple tissues [7] (fold enrichment = 1.2, $P < 1e-16$).”

In Results on page 10:

“For these eight mRNAs, we used cis-eQTL SNPs as IVs to test if the expression changes are causal for CVD traits by MR testing. At $P < 0.05/8$, we identified five genes whose expression was causal for CVD traits in whole blood using FHS-eQTLs (Supplementary Data 9), and five genes reflecting multiple tissues using GTEx eQTLs (Supplementary Data 10). Our results show for example that cg12555086, located in the LIPA gene body region, is associated in cis with the expression of LIPA ($\beta = -4.4$, and $P = 1e-277$). cis-meQTL variants associated with cg12555086 colocalized with cis-eQTL variants associated with of LIPA expression at a probability > 0.99 . LIPA expression tested causal for both CHD ($OR = 0.42$, and $P_{MR} = 1.4e-11$) and MI ($OR = 0.36$, and $P_{MR} = 6.1e-12$) in whole blood (Fig 6A-D). The expression levels of LIPA also tested casual for CHD and MI in many other tissues including adrenal gland, aorta, and liver. Another example is cg06882058 located in the gene body region of SDCCAG8; it is associated in cis with the expression of SDCCAG8 ($\beta = 1.72$, and $P = 1e-22$) and colocalized with cis-eQTL variants for SDCCAG8 (colocalization probability > 0.99). SDCCAG8 expression tested causal for both SBP ($\beta = 13.46$, and $P_{MR} = 7.2e-8$) and DBP ($\beta = 10.40$, and $P_{MR} = 1.6e-7$) in whole blood (Fig 6E-H). The expression levels of SDCCAG8 also tested casual for SBP and DBP in many other tissues including tibial artery, heart atrial appendage, aorta, and brain. These results highlight many putatively causal pathways for CVD traits involving both DNAm and gene expression changes. Further experimental validation is needed to definitively prove causality.”

7. Line 247: do you mean colocalized with SDCCAG8 eQTL?

Reply: We thank the Reviewer for this correction. We corrected the text as follows:

“Another example is cg06882058 located in the gene body region of SDCCAG8; it is associated in cis with the expression of SDCCAG8 ($\beta = 1.72$, and $P = 1e-22$) and colocalized with cis-eQTL variants for SDCCAG8 (colocalization probability > 0.99).”

8. Line 278: I would really like to see some kind of broad comparison between trans-meQTLs with trans-eQTLs to see if the conclusions are similar.

Reply: We previously performed a similar analysis by identifying *trans*-eQTL hotspots [17]. The *trans*-eQTL hotspots were defined by an index SNP associated with more than 10 transcripts. We found *trans*-eQTL hotspots that affect hundreds of genes, which was similar to what we now find for the *trans*-meQTL hotspots. However, the genes affected by *trans*-eQTLs hotspots did not show enrichment in transcriptional regulatory genes. Instead, the *trans*-eQTL hotspots were enriched for platelet SNPs and platelet eSNPs. In the revised the manuscript, we added the comparison of *trans*-meQTL hotspots with *trans*-eQTL hotspots (see the Discussion section, page 17):

“In our previous study, we identified 13 trans-eQTL hotspots that affected hundreds of genes [17]. The previously reported trans-eQTLs hotspots did not overlap with the trans-meQTL hotspots identified in the present study, and the cis-eGenes affected by trans-eQTL hotspots did not show enrichment of transcriptional regulatory genes. Instead, the trans-eQTL hotspots were enriched for platelet SNPs and platelet eQTL variants. This finding indicates that trans-eQTLs are highly tissue specific, in contrast to the trans-meQTLs, which reflect remote control by transcriptional regulatory genes.”

9. The discussion feels a little too specific on certain points. I would like to see a little bit more of conceptual ideas driving the end of the discussion section, especially around how coupling meQTLs and eQTLs can be advantageous for identifying disease-causal genes over simply using eQTLs and GWAS alone. The ending of the discussion feels abrupt with the limitations. I suggest folding the limitations throughout the discussion.

Reply: We thank the reviewer for this suggestion. We discussed the advantageous for identifying disease-causal genes by coupling meQTLs and eQTLs over simply using eQTLs and GWAS variants alone in the Discussion section. Please see our reply to Reviewer 3, Comment #1. We also folded in the limitations of our study throughout the Discussion section. Please see the revised Discussion section on page 12-17.

10. The figure labels for Figure 1B and 4A can be difficult to read when the labels overlap the bars. I suggest moving the labels above or below the bars

Reply: We moved the labels for Figure 1B and 4A below the bars.

11. Figure 7A may be better suited for the supplement.

Reply: We moved Figure 7A to the Supplementary Data, as Supplementary Figure 7.

12. I suggest moving the Molecular QTL browser section into the discussion or methods.

Reply: We moved the “Molecular QTL browser” section to “Data Availability” section after the “Methods” section.

13. I would like to see some pathway analysis on all of the putative causal genes identified in the manuscript beyond those that were just identified in the tran-meQTL results in the

Reply: We thank the reviewer for this suggestion. We used FUMA [18] to perform the pathway analysis on the meQTLs for the putatively causal CpGs. The results and methods are provided in the revised manuscript, as follows:

In the Methods section on page 26-27:

“Pathway analysis

To investigate possible pathways underlying the associations between meQTLs and CVD traits, we used FUMA[18] on the cis-meQTLs for the putatively causal CpGs identified by MR analysis. FUMA included all loci with cis-meQTLs at LD>0.8 based on the 1000 Genomes references panel as input.

Hypergeometric tests on genes from those loci were used to investigate over-representations of genes from multiple pathways. To improve focus in this study, we only use results of KEGG [19] and Gene Ontology – biological process (GO-BP) terms [20]. The SNP-to-Gene mapping was used on associations between SNPs and genes from positional mapping in Grch37/hg19, eQTLs in GTEx [7], and chromatin interaction mapping in Hi-C databases [21]. The significant threshold for the pathway analysis used a corrected $P < 0.05$ / tests pathways in FUMA.”

In the Results section on page 9:

“Pathways analysis by FUMA (Functional mapping and annotation of GWAS)[18] revealed that the cis-meQTLs for the 92 putatively causal CpGs for CVD traits were over-represented with genes involved in sterol metabolic process, regulation of lipoprotein lipase activity, and glycine, serine and threonine metabolism (Supplementary Data 9).”

References

1. Ghosh, S., et al., *Tissue specific DNA methylation of CpG islands in normal human adult somatic tissues distinguishes neural from non-neural tissues*. Epigenetics, 2010. **5**(6): p. 527-538.
2. Zhou, J., et al., *Tissue-specific DNA methylation is conserved across human, mouse, and rat, and driven by primary sequence conservation*. BMC genomics, 2017. **18**(1): p. 724.
3. Loh, K., et al., *DNA methylome profiling of human tissues identifies global and tissue-specific methylation patterns*. Genome biology, 2014. **15**(4): p. 3248.
4. Ding, J., et al., *Gene expression in skin and lymphoblastoid cells: Refined statistical method reveals extensive overlap in cis-eQTL signals*. The American Journal of Human Genetics, 2010. **87**(6): p. 779-789.
5. Flutre, T., et al., *A statistical framework for joint eQTL analysis in multiple tissues*. PLoS genetics, 2013. **9**(5): p. e1003486.
6. Kundaje, A., et al., *Integrative analysis of 111 reference human epigenomes*. Nature, 2015. **518**(7539): p. 317-330.
7. Consortium, G., *Genetic effects on gene expression across human tissues*. Nature, 2017. **550**(7675): p. 204.
8. Bowden, J., G. Davey Smith, and S. Burgess, *Mendelian randomization with invalid instruments: effect estimation and bias detection through Egger regression*. International journal of epidemiology, 2015. **44**(2): p. 512-525.
9. Sabater-Lleal, M., et al., *Genome-Wide Association Trans-Ethnic Meta-Analyses Identifies Novel Associations Regulating Coagulation Factor VIII and von Willebrand Factor Plasma Levels*. Circulation, 2019.
10. Verbanck, M., et al., *Detection of widespread horizontal pleiotropy in causal relationships inferred from Mendelian randomization between complex traits and diseases*. Nature genetics, 2018. **50**(5): p. 693.
11. Bonder, M.J., et al., *Disease variants alter transcription factor levels and methylation of their binding sites*. Nature genetics, 2017. **49**(1): p. 131.
12. Gaunt, T.R., et al., *Systematic identification of genetic influences on methylation across the human life course*. Genome biology, 2016. **17**(1): p. 61.
13. Lemire, M., et al., *Long-range epigenetic regulation is conferred by genetic variation located at thousands of independent loci*. Nature communications, 2015. **6**.
14. Giambartolomei, C., et al., *Bayesian test for colocalisation between pairs of genetic association studies using summary statistics*. PLoS genetics, 2014. **10**(5): p. e1004383.
15. Yao, C., et al., *Genome-wide mapping of plasma protein QTLs identifies putatively causal genes and pathways for cardiovascular disease*. Nature communications, 2018. **9**(1): p. 3268.
16. Pierce, B.L., et al., *Co-occurring expression and methylation QTLs allow detection of common causal variants and shared biological mechanisms*. Nature communications, 2018. **9**(1): p. 804.
17. Yao, C., et al., *Dynamic role of trans regulation of gene expression in relation to complex traits*. The American Journal of Human Genetics, 2017. **100**(4): p. 571-580.
18. Watanabe, K., et al., *Functional mapping and annotation of genetic associations with FUMA*. Nature communications, 2017. **8**(1): p. 1826.
19. Kanehisa, M. and S. Goto, *KEGG: kyoto encyclopedia of genes and genomes*. Nucleic acids research, 2000. **28**(1): p. 27-30.
20. Ashburner, M., et al., *Gene ontology: tool for the unification of biology*. Nature genetics, 2000. **25**(1): p. 25.
21. Lieberman-Aiden, E., et al., *Comprehensive mapping of long-range interactions reveals folding principles of the human genome*. science, 2009. **326**(5950): p. 289-293.

Reviewers' Comments:

Reviewer #1:

Remarks to the Author:

I think my comments have been accurately addressed with associated changes to the manuscript. This relates particularly to the limitation of being whole blood only. Thus I recommend it to be accepted with suggested changes.

Reviewer #2:

Remarks to the Author:

I thank the authors for addressing my comments. The manuscript is much improved, importantly the MR analyses look to be robust (though further clarification on P-value reporting is needed - see details below).

One remaining thought:

1. The authors consider the causal effects of methylation and gene expression on CVD traits and find 5 genes that do. For these genes, the authors have the opportunity to indicate whether the methylation or expression of these genes (or both) are driving the causal effects in an MR framework. This would also nicely illustrate the added value of meQTLs over and above eQTL.

Please can the authors clarify the following:

1. A mixture of both Bonferroni adjusted p-values and Bonferroni adjusted p-value thresholds (for interpretation of raw p-values) and unadjusted (raw) P-values are provided sometimes even within the same table e.g. SD9 provides: Bonferroni.pval.IVW.CpGtoTrait; Bonferroni.pval.IVW.mRNAtoTrait; log10P.eQTM. While in Table 1 no indication of whether the MR p-values are adjusted or not is given. Similarity in the main text, sometimes adjusted p-values, sometimes unadjusted P-values e.g. are the p-values on line 248 adjusted or unadjusted – perhaps this could be clarified? For SD9, perhaps the unadjusted P-values could be provided in addition to the adjusted? For main table 1, I think the unadjusted p-values for the MR analyses are provided (in contrast to the supplementary tables)? Similarly, when Bonferroni $P < 0.05$ are provided it would be useful for the reader to have a reminder about the number of tests that the P-values have been adjusted for. Having consistency and clarity on the P-values is pivotal for the reader to correctly interpret the findings.
2. Main table 1, please can the column headers be tidied up and the information provided clarified. "MR Pval" is this the IVW MR test P-value or the MR-Egger test P-value? Is it a Bonferroni adjusted P-value or unadjusted? When CHD/MI is listed as a trait, is the MI result provided or is the CHD results provided? Please correct the first * sentence for Table 1. Please also clarify whether the "* Bonferroni corrected P value is $0.05/14,910 = 3.35e-6$ " refers to an adjustment applied to the P-values in the table, or whether the P-values in the table should be interpreted using this as a threshold.
3. In the supplementary data files, can the content of the tables be described within the tables and particularly the column header information provided. The column headers are not always intuitive and therefore its challenging for a reader to interpret these data.
4. It would be helpful if the authors could tidy up the descriptions around the colocalisation analyses here. They only test under the one causal variant assumption yet at all these loci there are multiple IVs used for the test, hence the authors should re-phrase e.g. line 252 "A cis-meQTL variant associated with cg06882058 colocalized with a cis-eQTL variant associated with expression of SDCCAG8 (colocalisation probability > 0.99)" rather than phrasing as >1 variant colocalizing across traits. This should also be corrected elsewhere.
5. Line 117, the provided fold enrichment and P-value is identical to enhancer enrichment but in SF1 the fold enrichment looks less than 1.2.
6. Methods: line 525-529, it should be clarified that these aren't the latest GWAS for these traits. There is a much more recent CHD/MI GWAS by Pim van der Haarst for CHD for example, but more

importantly, some of these studies are metabochip studies and not GWAS at all (line 536).

Reviewer #3:

Remarks to the Author:

I am really appreciative of the authors' responses and have no further comments. I am satisfied with the additional analyses and changes to the text, and I believe the findings provide novel insights to the variant-to-gene mapping field and GWAS functional follow-up studies.

We are grateful for the additional comments from Reviewer #2. We carefully checked our manuscript again and corrected any inconsistencies in the paper. The changes are highlighted in the manuscript text file, and our detailed responses to Reviewers' questions are provided below.

In addition, we have ensured that our manuscript complies with the journal's policies and format requirements, and we have deposited our data in an enduring repository as described in the Data Availability section.

Detailed Responses to Reviewers' Comments

Reviewer #2 (Remarks to the Author):

One remaining thought:

1. The authors consider the causal effects of methylation and gene expression on CVD traits and find 5 genes that do. For these genes, the authors have the opportunity to indicate whether the methylation or expression of these genes (or both) are driving the causal effects in an MR framework. This would also nicely illustrate the added value of meQTLs over and above eQTL.

Reply: We overlooked the reviewer's request because we thought it was a summary comment. We now emphasize the value of combining meQTLs and eQTLs to infer causal pathways for human disease and traits. This is summarized in the Discussion on page 15, as follows:

“There is considerable merit for using meQTLs along with other molecular QTLs such as eQTLs to reveal much broader and more complex gene networks underlying genetic variant – disease associations. We show in the colocalization analyses that CpGs and their cis-associated gene expression are driven by the same causal variants. This suggests the presence of “vertical causal pathways” linking genetic variants, DNAm, and gene expression to human diseases.”

There were five genes for which both DNAm and gene expression tested causal for association with CVD risk factors. We have highlighted two examples in the Discussion section (page 15). The first example illustrates CpGs residing in known CVD-related genes (i.e., LIPA). Our results add evidence that DNAm is involved in causal pathways for CVD. The second example is for a CpG that tested positive by MR but was not previously reported to play a causal role in CVD (i.e., cg06882058 in SDCCAG8 in relation to systolic and diastolic BP). We have discussed these results in the Discussion (page 15) as follows:

“Some CpGs that tested positive by MR have not previously been reported to play a causal role in CVD. For example, we identified cg06882058 in SDCCAG8 as causal for both SBP and DBP. CpG cg06882058 is associated in cis with expression of SDCCAG8, which we found to be causal for both SBP and DBP (Supplementary Data 9-10). Previous studies found that SDCCAG8 causes nephronophthisis type 10, characterized by retinal and renal degeneration, mild intellectual disability, obesity, hypogonadism, and recurrent respiratory infections in humans^{46, 47}. Sdcccag8 knockout mice

develop late-onset nephronophthisis and severely increased BP^{d8}. This evidence leads us to hypothesize that dysregulation of cg06882058 may affect expression of SDCCAG8 and thereby cause hypertension and contribute to CVD risk. Further experimental validation is necessary to prove this hypothesis.”

Please can the authors clarify the following:

1. A mixture of both Bonferroni adjusted p-values and Bonferroni adjusted p-value thresholds (for interpretation of raw p-values) and unadjusted (raw) P-values are provided sometimes even within the same table e.g. SD9 provides: Bonferroni.pval.IVW.CpGtoTrait; Bonferroni.pval.IVW.mRNAtoTrait; log10P.eQTM. While in Table 1 no indication of whether the MR p-values are adjusted or not is given. Similarity in the main text, sometimes adjusted p-values, sometimes unadjusted P-values e.g. are the p-values on line 248 adjusted or unadjusted – perhaps this could be clarified? For SD9, perhaps the unadjusted P-values could be provided in addition to the adjusted? For main table 1, I think the unadjusted p-values for the MR analyses are provided (in contrast to the supplementary tables)? Similarly, when Bonferroni P<0.05 are provided it would be useful for the reader to have a reminder about the number of tests that the P-values have been adjusted for. Having consistency and clarity on the P-values is pivotal for the reader to correctly interpret the findings.

Reply: We thank the reviewer for this suggestion. To be consistent, in the revised manuscript, we now provide both raw P-values and adjusted P values in Table 1, and Supplementary Data S6, S9, and S13. In the main text, we now provide Bonferroni corrected P-values and the number of tests for correction. These changes are highlighted in the revised manuscript.

2. Main table 1, please can the column headers be tidied up and the information provided clarified. “MR Pval” is this the IVW MR test P-value or the MR-Egger test P-value? Is it a Bonferroni adjusted P-value or unadjusted? When CHD/MI is listed as a trait, is the MI result provided or is the CHD results provided? Please correct the first * sentence for Table 1. Please also clarify whether the “* Bonferroni corrected P value is 0.05/14,910=3.35e-6” refers to an adjustment applied to the P-values in the table, or whether the P-values in the table should be interpreted using this as a threshold.

Reply: We have revised the column headers in Table 1 as suggested. Both the raw P-value and Bonferroni corrected P-values have been added. The new headers are now as follows:

CpG	Phenotype	Chr	Gene	Number of Independent cis-meQTLs	IVW MR test OR	IVW MR test 95% CI	IVW MR test P-value	IVW MR test Bonferroni corrected P-value	Heterogeneity test P-value	Pleiotropy test P-value
-----	-----------	-----	------	----------------------------------	----------------	--------------------	---------------------	--	----------------------------	-------------------------

We also revised the *sentence under Table 1 as below.

** For CpGs that tested causal for both MI and CHD, only the MR results for CHD are shown in this table. The full MR results are reported in Supplementary Data 6.*

** The Bonferroni corrected P value is corrected for the number of CpGs having ≥ 3 independent cis-meQTLs (N=14,910).*

** Independent cis-meQTLs were defined using LD $r^2 < 0.01$*

3. In the supplementary data files, can the content of the tables be described within the tables and particularly the column header information provided. The column headers are not always intuitive and therefore its challenging for a reader to interpret these data.

Reply: We have revised column headers in Supplementary Data S5, S6, S7, S8, S9, S10, S13, and S14 to make them self-explanatory.

4. It would be helpful if the authors could tidy up the descriptions around the colocalisation analyses here. They only test under the one causal variant assumption yet at all these loci there are multiple IVs used for the test, hence the authors should re-phrase e.g. line 252 “A cis-meQTL variant associated with cg06882058 colocalized with a cis-eQTL variant associated with expression of SDCCAG8 (colocalisation probability > 0.99)” rather than phrasing as >1 variant colocalizing across traits. This should also be corrected elsewhere.

Reply: We thank the reviewer for this suggestion. We have corrected the description for colocalization, on line 246, 252, and the figure legends of figures 6B and 6D as follows:

Line 246:

“A cis-meQTL variant associated with cg12555086 colocalized with a cis-eQTL variant associated with LIPA expression at a probability >0.99.”

Line 252:

“A cis-meQTL variant associated with cg06882058 colocalized with a cis-eQTL variant associated with expression of SDCCAG8 (colocalization probability >0.99).”

Figure legends for Figure 6:

“Figure 6: Mendelian randomization examples. A) MR example of cg12555086 in relation to CHD and MI; B) colocalization of a casual cis-meQTL and a casual cis-eQTL on cg12555086 and LIPA expression; C) MR example of cg06882058 in relation to SBP and DBP; D) colocalization of a casual cis-meQTL and a casual cis-eQTL on cg06882058 and SDCCAG8 expression.”

5. Line 117, the provided fold enrichment and P-value is identical to enhancer enrichment but in SF1 the fold enrichment looks less than 1.2.

Reply: The text in line 117 has been revised as follows:

“There were 6,212 CpGs with household effects >0.1 (Supplementary Data 4) that showed enrichment for location in 3’UTR regions (fold enrichment =1.26, P<1e-16) and depletion in promoter, TSS200, CpG Islands, and high-CpG dense regions (fold enrichment <0.8, P<1e-16, Supplementary Fig 1).”

6. Methods: line 525-529, it should be clarified that these aren’t the latest GWAS for these traits. There is a much more recent CHD/MI GWAS by Pim van der Haarst for CHD for example, but more importantly, some of these studies are metabochip studies and not GWAS at all (line 536).

Reply: We have clarified these limitations in the Methods section on page 22 and as follows:

“A large proportion of meQTLs were not measured on the Metabochip genotyping platform, which limited MR analyses for BMI and lipid traits. When the manuscript was under preparation, a new GWAS for CHD was published⁷¹ that was not incorporated in the MR analysis.”